# The role of temporal scales in extracting dominant meteorological drivers of major airborne pollutants

Miaoqing Xu[1,2], Jing Yang[1], Manchun Li[3], Xiao Chen[1], Qiancheng lv[1], Qi Yao[1], Bingbo Gao[4], Ziyue Chen[1]*

[1]Faculty of Geographical Science, Beijing Normal University, 19 Xinjiekou Street, Haidian, Beijing 100875, China
[2]Hubei Provincial Academy of Eco-environmental Sciences (Hubei Eco-environmental Engineering Assessment Center), Wuhan 430070, China
[3]School of Geography and Ocean Science, Nanjing University, Nanjing 210023, China
[4]College of Land Science and Technology, China Agricultural University, Beijing 100083, China

*Correspondence to*: Ziyue Chen (zychen@bnu.edu.cn)

**Abstract.** The influence of individual meteorological factors on different airborne pollutants has been massively conducted. However, few studies have considered the effect of temporal scales on the extracted pollutant-meteorology association. Based on Convergent Cross Mapping (CCM), we compared the influence of major meteorological factors on $PM_{2.5}$, $PM_{10}$ and $O_3$ concentrations in 2020 at the 3h and 24h scale respectively. In terms of the extracted dominant meteorological factor, the consistence between the analysis at 3h and 24h scale was relatively low, suggesting a large difference even from a qualitative perspective. In terms of the mean ρ value, the effect of temporal scale on PM ($PM_{2.5}$ and $PM_{10}$)-Meteorology association was consistent, yet largely different from the temporal-scale effect on $O_3$. Temperature was the most important meteorological factor for $PM_{2.5}$, $PM_{10}$ and $O_3$ across China at both 3h and 24 scale. For $PM_{2.5}$ and $PM_{10}$, the extracted PM-temperature association at the 24h scale was stronger than that at the 3h scale. Meanwhile, for summer $O_3$, due to strong reactions between precursors, the extracted $O_3$-temperature association at the 3h scale was much stronger. Due to the discrete distribution, the extracted association between all pollutants and precipitation was much weaker at the 3h scale. Similarly, the extracted PM-wind association was notably weaker at the 3h scale. Due to precursor transport, summertime $O_3$-wind association was stronger at the 3h scale. For atmospheric pressure, the pollutant-pressure association was weaker at the 3h scale except for summer, when interactions between atmospheric pressure and other meteorological factors were strong. From the spatial perspective, pollutant-meteorology associations at 3h and 24h were more consistent in those heavily polluted regions, while extracted dominant meteorological factors for pollutants demonstrated more difference at 3h and 24h in those less polluted regions. This research suggested that temporal scales should be carefully considered when extracting natural and anthropogenic drivers for airborne pollution.

# 1 Introduction

Since 2013, PM$_{2.5}$ induced haze events increased dramatically across China (Chen et al., 2020a; Wang et al., 2021a). To address this issue, a series of emission-cut policies were released and strictly implemented, leading to significantly reduced PM$_{2.5}$ concentrations at the national scale (Wang et al., 2021b; Wang et al., 2022; Xiao et al., 2020). Conversely, with the improvement of PM$_{2.5}$ pollution, a soaring ground ozone level was observed since 2013, making composite airborne pollution a rising challenge (Gong et al., 2017; Zheng et al., 2018; Nelson et al., 2021). Against this background, a comprehensive

understanding of their characteristics and driving factors is key for effectively predicting and managing composite airborne pollutants (Chen et al., 2018, 2019a, 2019c, 2020a).

The major influential factors for airborne pollutants are human factors, which closely relates to their compositions and formation (Cheng et al., 2017; Zhan et al., 2017), and meteorological factors, which closely relates to their dispersion (Chen

et al., 2020; Guo et al., 2020; Zhang et al., 2020). Given the strong negative effects of airborne pollution on public health (Kelly et al., 2015; Gao et al., 2017; Yin et al., 2020) and crop yields (Zhou et al., 2018; Xu et al., 2021), massive studies have been conducted on the human and meteorological attribution of composite airborne pollution. For meteorological influencing factors, Yang et al. (2021) studied 284 major cities in China based on daily scales and found that PM$_{2.5}$ was mainly affected by wind, temperature and rainfall, while O$_3$ was mainly affected by temperature, relative humidity and sunshine duration.

Wang et al. (2018) established 12 joint regression models and analyzed that the leading meteorological factors of PM$_{2.5}$ pollution in Zhejiang were temperature and wind speed based on the hour-scale data. For emission influencing factors, Wang et al. (2018) found that the emission influencing factor of PM$_{2.5}$ pollution in Zhejiang was NO$_2$ based on the analysis of hour-scale data. Zhai et al. (2019) estimated the correlation between PM$_{2.5}$ concentration and meteorological factors at the 10-day scale found that the variation trend of PM$_{2.5}$ and SO$_2$, NO$_2$ and CO was consistent, and SO$_2$ emission-control was the main

driving factor for PM$_{2.5}$ variations. In addition to the variation of seasons and geographical locations, the temporal resolution of data sources can be another major reason for the distinct outputs. Fu et al. (2020) used integrated empirical mode decomposition (EEMD) to decompose the time series data of PM$_{2.5}$, five other atmospheric pollutants and six meteorological types. On the daily scale, PM$_{2.5}$ was positively correlated with O$_3$ and daily maximum and minimum temperature, and negatively correlated with air pressure, while PM$_{2.5}$ presented an opposite association with these factors at the monthly scale.

Despite massive studies conducted, notable inconsistence of dominant meteorological and anthropogenic drivers for airborne pollutants was observed between findings from previous studies. Even if some studies revealed different pollutant-meteorology association at multiple temporal scales, such research conducted in isolated cities, cannot reflect the spatiotemporal variations of temporal effects across China. More importantly, due to the lack of high temporal-resolution data, previous studies were mainly conducted at the daily scale, while many scholars may believe that the application of high-temporal-resolution data

leads to a better extraction of pollutant-meteorology association.

To fill this gap, we employed the data of major airborne pollutants, including $PM_{2.5}$, $PM_{10}$ and $O_3$, meteorological factors and some precursors across China with a temporal resolution of 3h and 24h respectively. By comparing the major drivers for airborne pollutants extracted using data sources with different temporal resolution, the role of temporal scales in the attribution of composite airborne pollution can be comprehensively understood. This research aims for an improved understanding of the mechanisms how different factors may affect airborne pollutants under various temporal scales and sheds useful light on a better management of composite airborne pollution through more effective emission-cut measures.

## 2 Methodology

### 2.1 Data sources

3 hourly meteorological data across China for January- December 2020 were obtained from the China Meteorological Administration. The meteorological variables used in this study included temperature, precipitation, wind direction, wind speed and atmospheric pressure, which were closely related to $PM_{2.5}$, $PM_{10}$ (Chen et al., 2020a) and $O_3$ concentrations (Chen et al., 2020b). For cities with more than one observation station, the average of records from multiple stations was employed. For a multi-scale comparison, the 24h meteorological data were produced by conducting average operation on the 3h meteorological data. The previous studies have proved that pollutant-meteorology association presented notable seasonal variations and thus if CCM has been conducted based a whole year, the p value was not significant at many cases and thus the comparison cannot be conducted. Therefore, in this research, we considered the experiments based on seasonal data respectively. For analyzing seasonal variations of pollutant-meteorology association, December, January, February were set as winter, March, April, May as Spring, June, July, August as Summer, September, October, November as Autumn.

Hourly concentration data of $PM_{2.5}$, $PM_{10}$ and $O_3$ during the same period were obtained from China National Environmental Monitoring Center, CNEMC. The meteorological data were matched according to cities and air pollutant stations, and the nearest station corresponding to each air pollution monitoring station was selected as its surrounding meteorological conditions. A total of 101 cities were successfully matched. For cities with more than one observation station, the average of records from multiple stations was employed. To match the temporal scale of meteorological data, the per-3h and per-24h pollutant data were produced by conducting average operation on the hourly concentration data.

### 2.2 Advanced Causation Model

Since 2013, when $PM_{2.5}$ pollution was observed across China, research on airborne pollution has been massively conducted. Amongst a diversity of topics, research on the meteorological influences on major airborne pollutants (e.g., $PM_{2.5}$ and $O_3$) has received growing emphasis. However, the major challenge for extracting and comparing the influence of individual meteorological factors lies in the complex inner-interactions between multiple meteorological factors, which cause large uncertainties when applying traditional correlation analysis (Chen et al., 2020a). To address this issue, we employed an

advanced causation model, Convergent Cross Mapping (CCM), to quantify the influence of each meteorological factor on $PM_{2.5}$, $PM_{10}$ and $O_3$. By removing the influence of disturbing factors, CCM (Sugihara et al., 2012) is capable of extracting

reliable coupling between two variables in complex ecosystems. CCM calculates the causal influence of the variable A on the target variable B as the ρ value, ranging from 0 to 1. Like the correlation coefficient, the ρ value can be used for comparing the influencing between multiple variables on the target variable.

Thanks to its advantage in effectively extracting the asymmetric, bidirectional association between two variables and

100 identifying mirage correlation in complex ecosystems with a diversity of variables, we have massively employed CCM to evaluate the influence of multiple meteorological factors on $PM_{2.5}$ (Chen et al., 2017, 2018), $O_3$ (Cheng et al., 2019; Chen et al., 2020b) and NPP (Gao et al., 2022) and achieved reliable outputs. Based on a multi-model comparison experiment, our recent research (Chen et al., 2022) proved that CCM was the most suitable model for causation inference in complex atmospheric environment. CCM is specifically designed to deal with the nonlinear relationship between two variables and is

105 fully suitable for the nonlinear relationship between atmospheric factors. Compared with other mainstream statistical models, CCM was advantageous of identifying unique pollutant-meteorology association in local areas while maintaining general characteristics of pollutant-meteorology association across China. Furthermore, CCM generated meteorology-pollutant association were highly consistent with prior-knowledge. In this regard, for this research, we also employed CCM to quantify and compare the influence of temperature, precipitation, wind speed, wind direction and atmospheric pressure on $PM_{2.5}$, $PM_{10}$

and $O_3$ concentrations. CCM automatically considers all possible interaction forms and lag effects between the time series of two variables, which effectively reduces the influence of interference and avoids the influence of other factors. CCM is highly automatic to remove the uncertainty of manual setting and only the setting of several parameters is required: E (number of dimensions), τ (time lag) and b (number of nearest neighbors). For this research, τ, E, and b were set as 2 days, 3 and 4 according to previous studies (Chen et al., 2018; 2020b).

Based on the rarely employed 3h meteorological data sources, we compared the effects of temporal scales on the extracted pollutant-meteorology causation. Acknowledged, due to the data limitation at the 3h scale, which did not include humidity and sunshine duration, we could simply consider a limited number of meteorological factors (Temperature, Precipitation, Wind Speed, Wind Direction and Atmospheric Pressure), which was less than our previous studies based on meteorological data at

120 the 24h scale, while some meteorological factors (e.g., humility and sunshine duration) were missed in this research. However, since we compared the same set of these major meteorological factors at both 3h and 24h scale, the calculated consistence and difference could effectively reveal the potential effects of different temporal scales on the quantitative (the detailed ρ value) and qualitative (the dominant meteorological factor) findings of pollutant-meteorology association. Despite the limitation of number of meteorological factors, it caused limited influence on the temporal effects on pollutant-meteorology association.

This is because CCM simply considers the causality between the target variable and one influencing variable, and removes the influence from other variables (Sugihara et al., 2012; Chen et al., 2020). Another limitation of this data was that this data set

simply included one year's data and thus the inter-annual variation of temporal effects on pollutant-meteorology association could not be revealed. For this research, we majorly revealed the existence of strong temporal effects on pollutant-meteorology association, which can be fully supported by the one-year data with four seasons (four complete time series with more than 90

records (24h scale) and 720 records (3h scale)Meanwhile, the temporal variation of temporal effects on pollutant-meteorology association and its influencing factors should be further investigated in future studies, when the long time series data sets of 3h meteorological data become available.

**3 Results**

**3.1 The comparison of dominant meteorological factors for $PM_{2.5}$, $PM_{10}$ and $O_3$ across China at 3h and 24h scale**

Based on CCM, we calculated the dominant meteorological factors for seasonal $O_3$ (Table 1), $PM_{2.5}$ (Table 2) and $PM_{10}$ (Table 3) concentrations at the 3h and 24h respectively. By comparing the extracted pollutant-meteorology association, we calculated the number of cities with the same meteorological factor at different temporal scales (Table 4). As shown in Table 4, the consistence between dominant meteorological factors for $PM_{2.5}$, $PM_{10}$ and $O_3$ at two temporal scales varied significantly

(ranging from 31.68% ~ 61.29%), indicating the temporal scale played a large role in the analysis of pollutant-meteorology association. As can be seen from *Table 1, Table 2 and Table 3,* from the seasonal scale, the consistence between dominant meteorological factors extracted at 3h and 24h in autumn and winter was higher than that in spring and summer. For example, temperature, precipitation, etc., $O_3$, $PM_{2.5}$, and $PM_{10}$ were mostly more dominant in autumn and winter than in spring and summer. This phenomenon indirectly suggested that meteorological influences on airborne pollutants were stronger in autumn

and winter, and thus the role of dominant meteorological factor was highlighted.

Table 1 inserted here.

Table 2 inserted here.

Table 3 inserted here.

As can be seen from *Table 1,* at the 3h scale, for $O_3$, the number of cities with precipitation as the dominant influencing factor was largest in winter, with 43 cities, while the number of cities with temperature was largest in spring, summer and autumn,

with 64 cities, 78 cities, and 75 cities, respectively; As one can see from *Table 2 and Table 3,* for $PM_{2.5}$ and $PM_{10,}$ the number of cities with temperature was largest in all seasons. As a comparison, at the 24h scale, for $O_3$, the number of cities with temperature as the dominant influencing factor was largest in spring, with 59 cities, and for $PM_{2.5}$ and $PM_{10}$, the number of

cities with temperature as the dominant influencing factor was largest in autumn, with 61, 55 cities, respectively, which was consistent with previous studies (Wang et al., 2018; Yang et al., 2021), while the number of cities with precipitation was largest in winter, with 47, 35, and 36 cities, respectively.

Table 4 inserted here.

However, the consistence of dominant factors between two temporal scales remained less than 50%. The study identified the dominant meteorological factors through CCM according to the $\rho$ value. While $\rho$ of the dominant meteorological factor was largest, it may be just slightly larger than $\rho$ of other meteorological factors at 24h (3h) scale, and may be smaller than $\rho$ of another factor, which led to the change of dominant factor, at 3h (24h) scale. In this case, if we simply consider the difference between the dominant meteorological factor (with the largest $\rho$) at 3h and 24h scale, the analysis was qualitative and not sufficient, which cannot comprehensively reveal the difference of pollutant-meteorology association at different temporal scales. Therefore, we further analyzed the detailed $\rho$ for all meteorological factors on $O_3$, $PM_{2.5}$ and $PM_{10}$ at two temporal scales respectively, to present a quantitative and comprehensive comparison.

### 3.2 The comparison of quantified influence of different meteorological factors on $PM_{2.5}$, $PM_{10}$ and $O_3$ across China at 3h and 24h scale

The detailed distribution of influence of individual meteorological factors on $O_3$, $PM_{2.5}$ and $PM_{10}$ concentrations are presented in Figure 1. Generally, meteorological influences on airborne pollutants presented a consistent trend between the 3h and 24h scale, characterized with a generally similar violin shape. According to Figure 1, the violin shape and range of 3h pollutant-meteorology was much sharper than the 24h pollutant-meteorology, indicating the 3h temporal scale was more sensitive to reveal the variation of pollutant-meteorology interactions. As shown in Table 5, similar to the number of dominant meteorological factors, the mean of calculated $\rho$ value across China also proved that temperature exerted a much stronger influence on $PM_{2.5}$, $PM_{10}$ and $O_3$ than other factors. Furthermore, according to the violin shape of different pollutants, we found that the pattern of $PM_{2.5}$-Meteorology and $PM_{10}$-Meteorology was generally consistent and largely different from the pattern of $O_3$-Meteorology, indicating that meteorological influences on particulate matters and gaseous pollutants were different. The major difference of pollutant-meteorology interactions at 3h and 24h was explained as follows:

Table 5 inserted here.

Figure 1 inserted here.

For all three airborne pollutants, temperature exerted a strongest influence across China in all seasons in terms of the largest mean $\rho$. High temperature promotes photochemical reactions and produce more $PM_{2.5}$, $PM_{10}$ and other precursors of secondary

pollutants, leading to higher concentrations of PM$_{2.5}$ and PM$_{10}$. High temperature may also lead to increased evaporation loss of PM$_{2.5}$ and PM$_{10}$, including NO$^{3-}$ salt and other volatile or semi-volatile components, resulting in decreased concentrations of PM$_{2.5}$ and PM$_{10}$. For PM$_{2.5}$ and PM$_{10}$, the calculated influence of temperature at 24h scale was consistently larger than that at the 3h scale. This may be attributed to the fact that the secondary reactions of the precursors of PM were less-intensive (Chen et al, 2016, 2020) and thus the temperature variation within 24h exerted a stronger influence than 3h temperature variation. Meanwhile, the influence of temperature on O$_3$ presented a notable seasonal pattern. For the relatively cold season winter and spring, when O$_3$ concentrations were relatively low, the influence of temperature at 24h scale was larger than that at the 3h scale. For summer, when O$_3$ concentrations were the highest, the influence of temperature at 3h scale was much larger than that at the 24h scale. This is mainly attributed to the fact that the high temperature in summer was the major trigger for quick reactions between precursors and high O$_3$ concentrations. Therefore, short-term variations of temperature could strongly influence O$_3$ concentrations in summer (Cheng et al., 2018, 2019).

For precipitation, since the distribution of precipitation in a day's time is not unified, and there may be no precipitation in many 3h slots, the mean ρ of precipitation across China at the 3h scale was weaker than that at the 24h scale. As a comparison, at the 24 scale, the occurrence of precipitation was significantly enhanced and thus the influence of precipitation on airborne pollutants was much stronger. Across China, the precipitation intensity showed obvious seasonal variations, and most regions may have the maximum value in summer and minimum value in winter. The eastern region of China is affected by summer monsoon in summer and autumn, there is a lot of precipitation; In winter, China receives less precipitation due to the influence of winter winds.   Accordingly, the calculated ρ of precipitation on PM$_{2.5}$, PM$_{10}$ and O$_3$ at 24h scale was remarkably larger than that at 3h scale in summer.

Previous studies (Chen et al., 2017, 2018, 2020) proved that wind played a notable influence on PM. Similar to precipitation, the daily distribution of wind is not unified, and there may be calm wind conditions in many 3h slots. Therefore, the mean ρ of wind direction and wind speed on PM$_{2.5}$ and PM$_{10}$ at 24h scale was notably larger than that at the 3h scale. Wind-O$_3$ interactions presented notable seasonal patterns. In the less-polluted Spring and Winter, the mean ρ of wind direction and wind speed at the 24h scale was larger than that at the 3h scale. In summer, when O$_3$ concentrations were relatively high, the mean ρ of wind direction and wind speed at the 3h scale was larger.

Atmospheric pressure mainly affects the transport and accumulation of pollutants by indirectly influencing other meteorological factors (e.g. wind and precipitation). Therefore, large uncertainties existed in the extracted pressure-pollutant causation. Generally, for PM$_{2.5}$, PM$_{10}$ and O$_3$, the mean ρ of atmospheric pressure across China at the 3h scale was weaker than that at the 24h scale, except for summer, when the interactions between atmospheric pressure and other meteorological factors were strong.

**3.3 The spatial patterns of dominant meteorological factors for PM$_{2.5}$, PM$_{10}$ and O$_3$ across China at 3h and 24h scale**

As shown in Figure 2, all the locations of the mentioned regions have been marked. As shown in Figure 3, 4, 5, the influence of meteorological factors on airborne pollutants has obvious seasonal variations and presented some regional similarity. The seasonal concentration of air pollutant data for each city is calculated using the average of hourly concentration data measured by all available local observation stations. For PM$_{2.5}$(Figure 3)and PM$_{10}$(Figure 4), the dominant meteorological factor for Northeast China was mainly wind, especially the heavily polluted winter, while the dominant meteorological factor for Yangtze

River Delta was mainly precipitation at both 3h and 24h scale. The dominant meteorological factor in Shandong Peninsula in spring and autumn, southern China in summer, northern and coastal areas in autumn, and northeast China in winter are also consistent at different temporal scales. For O$_3$(Figure 5), especially the heavily polluted summer, temperature presented a prevailing role across the nation and was the dominant role for most cities. This output was consistent with our previous studies (Chen et al., 2018, 2019a), suggesting the general national trend of Pollutant-Meteorology association varied limitedly across

temporal scales of research data, especially in those heavily polluted regions. Meanwhile, for those regions, where the airborne pollution was not severe and homogeneous, the temporal issues of meteorological influences on PM was notable and thus the dominant meteorological factor in these regions presented notable differences at 3h and 24h scale.

Based on the extracted pollutant-meteorology associations at 3h scale, which have rarely been discussed, we found some

interesting differences of pollutant-meteorology association between 3h and 24h in some major regions across China.  For the heavily polluted Beijing-Tianjin-Hebei Region, the dominant meteorological factor for O$_3$ in spring was temperature at 3h scale. Meanwhile, the dominant factor was wind speed at the 24h scale. For PM$_{2.5}$, the dominant factor for PM$_{2.5}$ in spring was temperature at the 3h scale and wind speed at the 24h scale. The dominant meteorological factor for PM$_{10}$ in summer was temperature at the 3h scale and precipitation at 24h scale.

For the Yangtze River Delta, the dominant meteorological factor for O$_3$ in spring was temperature at the 3h scale and the combination of temperature and precipitation at the 24h scale. In summer, the dominant meteorological factor for O$_3$ was temperature at the 3h scale and wind speed at the 24h scale. The dominant factor of PM$_{2.5}$ in spring was temperature at the 3h scale and the combination of temperature and precipitation at the 24h scale. The dominant factor of PM$_{10}$ in spring was mainly

temperature at the 3h scale and wind speed at the 24h scale. For the Pearl River Delta, the dominant meteorological factor for O$_3$ in winter was temperature at the 3h and precipitation at the 24h scale.

For Sichuan Basin, the dominant meteorological factor for O$_3$ in all four seasons was constantly temperature at the 3h time scale, while it was precipitation, atmospheric pressure and wind speed in summer, autumn and winter respectively at the 24h

scale. The dominant meteorological element for PM$_{2.5}$ was temperature in all four seasons at the 3h scale, while it was precipitation in summer and winter at the 24h scale. The dominant meteorological element for PM$_{10}$ in spring and winter was

temperature at the 3h scale,   while it was atmospheric pressure for spring and winter at the 24h scale. Compared with other regions, the unique basin terrain led to stronger temporal effects on extracted pollutant-meteorology associations.

Our previous (Chen et al., 2018; 2020) revealed that meteorological influences exerted a stronger influence on PM pollutants when PM concentration was higher. This might be the reason that the difference of PM-meteorology associations between 3h and 24h was relatively small in heavily polluted winter and large in less-polluted spring. Meanwhile, we found that the role of wind speed and precipitation may be largely underestimated at the 3h scale. Compared with the generally consistent pollutant-meteorology associations in these heavily polluted regions, the dominant factor for PM and $O_3$ demonstrated significant

variations in those coastal cities, such as Shenzhen, Zhuhai, Zhanjiang.

Figure 2 inserted here.

Figure 3 inserted here.

Figure 4 inserted here.

Figure 5 inserted here.

**4 Discussion**

Although previous studies (Tai et al., 2010; Hu et al., 2021b; Yousefian et al., 2021; Zhong et al., 2021) pointed out the notable differences of pollutant-meteorology associations at different temporal scales and the great importance to better understand the temporal effects, few studies actually conducted a comparative analysis due to the lack of data, especially the high temporal resolution meteorological data. This research suggested that the temporal effects on pollutant-meteorology association was

significantly strong. While the obvious quantitative difference of the influence of individual factors on $PM_{2.5}$, $PM_{10}$ and $O_3$ (as shown in Figure 1), we found a very low consistence between extracted dominant meteorological factors (the consistence was less than 50% for all pollutants), indicating strong temporal effects even from a qualitative perspective. Based on the comparison of extracted pollutant-meteorology association at the 3h and 24h, there were no fixed spatiotemporal patterns of pollutant-meteorology association across temporal scales. However, we got some major conclusions as follows. Firstly, we

found the temporal effects of meteorological influences on different PM (e.g., $PM_{2.5}$ and PM) were similar, yet notably different from that on gaseous pollutants (e.g., $O_3$). Secondly, there were notable differences of the temporal effects between different meteorological factors. The variation of pollutant-meteorology association for those factors with continuous observation record (e.g., Temperature) was notably different from those factors with discrete observation record (e.g. Precipitation) at 3h and 24h

scale. The role of wind speed and precipitation, which may be recognized as dominant meteorological factors at the 24h scale, can be largely underestimated at 3h scale. Thirdly, the effects of temporal scales on pollutant-meteorology association varied significantly across seasons, characterized with notable difference between heavily polluted and less polluted seasons (e.g. The heavily polluted season for $O_3$ and PM was summer and winter respectively). Despite a complicate pattern, we found that the heavier the pollution, the stronger pollutant-meteorology association was. Consequently, in the heavily polluted season, the short-term (e.g., 3h) variation of specific meteorological factors (e.g. Temperature, Wind speed) exerted a stronger influence on PM and $O_3$ than the daily variation. The concentrations of PM and $O_3$ largely depend on wind conditions. High $O_3$ concentrations in different cities usually occur in the presence of strong wind speed, but are independent of wind direction, while high PM is often accompanied by weak wind speed, poor dispersion conditions, and sometimes occurs in strong northerly or southerly winds. The regional transport of air pollutants between cities is common (Li et al., 2019). As a comparison, in the less polluted season, the daily accumulation of specific meteorological factors exerted a stronger influence on airborne pollutants than short-term (e.g., 3h) accumulation. While the general trend of pollutant-meteorology association was consistent with previous studies, the general ρ value was slightly smaller for this research. The underlying reason may be the reduced $PM_{2.5}$ concentration in 2020 caused by the emission-cut during COVID-19. As explained in our previous studies (Chen et al., 2018), the higher $PM_{2.5}$ concentration, the stronger meteorological influence on $PM_{2.5}$ concentrations. Similar to our previous studies (Chen et al., 2017, 2018, 2022), we conducted the CCM analysis at the seasonal scale. This is because the large seasonal variation of pollutant-meteorology association may cause an insignificant output of CCM for an entire-year analysis, and cause large uncertainties.

This research suggested that the temporal scale played a complex role and higher temporal-resolution did not guarantee a stronger pollutant-meteorology association. For instance, for hot seasons (e.g., summertime $O_3$), the reaction between $O_3$ precursors was strong and quick, and thus the 3h resolution could better feature the influence of temperature on $O_3$ concentrations. Meanwhile, the secondary reaction for $PM_{2.5}$ was relatively slow (Chen et al. 2016), and the daily variation of temperature and $PM_{2.5}$ concentrations presented a stronger association than the hourly variation of temperature and $PM_{2.5}$ concentrations. Similarly, due to the discrete distribution, the daily influence of daily total precipitation on daily $PM_{2.5}$ concentrations was also notably stronger than the influence of 3h precipitation on 3h $PM_{2.5}$ concentrations. Furthermore, this type of uncertainty was not predicable across regions. Given the complex effects of temporal scales on pollutant-meteorology association, scholars should properly choose the temporal-resolution of research data according to the aims, study sites, pollutant types and seasons. With the growing availability of long-term meteorological and pollutant data, multi-scale, instead of high-temporal-resolution, research is recommended to comprehensively understand the short- and long-term meteorological influences on different airborne pollutants.

For future research, the temporal effects of influence of meteorological factors (e.g., Humidity, Boundary layer height) on airborne pollutants should also be explored with the availability of new data sources. On the other hand, this research proved

the important role of temporal scales in quantifying the influence of meteorological factors on airborne pollutants. Similarly, when inferring the association between precursors ($NO_2$, VOCs) and airborne pollutants, the temporal scales, which was rarely considered in previous studies, should also be comprehensively taken into account. The reaction rate between different precursors and the target pollutants in different regions and seasons could be better understood through multi-scale causation analysis. CCM is an ideal tool for quantifying the influence of individual meteorological factors on $PM_{2.5}$ concentrations, as it can effectively remove the influence of other meteorological factors. Therefore, this research revealed a strong temporal effect on pollutant-meteorology association, from the perspective of the association of individual meteorological factors. However, admittedly, CCM is limited in establishing the overall effects of multiple meteorological factors on $PM_{2.5}$ concentrations. Instead, other models, such as GAM (Generalized Additive Model), which work limited in extracting the association between $PM_{2.5}$ and individual meteorological factors, are advantageous in extracting the overall influence of multiple meteorological factors on airborne pollutants (Gong et al., 2017; Zheng et al., 2018; Hu et al., 2021a). When such 3h meteorological data set become more easily available and includes a complete set of meteorological factors, we could also employ GAMs or CTMs to investigate the temporal effects on the combined effects of meteorological factors on airborne pollutants.

**5 Conclusion**

We employed CCM to compare the influence of major meteorological factors (Temperature, Precipitation, Wind Speed, Wind Direction and Atmospheric Pressure) on $PM_{2.5}$, $PM_{10}$ and $O_3$ concentrations in 101 cities across China at the 3h and 24h scale in 2020. Results revealed a strong effect of temporal scale on the pollutant-meteorology association from different perspective. In terms of the extracted dominant meteorological factor, the consistence between the analysis at 3h and 24h scale was relatively low (the consistence for all pollutants was less than 50%), suggesting a large difference even from a qualitative perspective. In terms of the mean ρ value, the effect of temporal scale on the influence of individual meteorological factors on Particulate Matter ($PM_{2.5}$ and $PM_{10}$) was consistent, which was largely different from the temporal-scale effect on gaseous pollutants. Temperature was the most important meteorological factor for $PM_{2.5}$, $PM_{10}$ and $O_3$ across China at both 3h and 24 scale. For $PM_{2.5}$ and $PM_{10}$, the secondary reaction was less-intense, the extracted PM-temperature association at the 24h scale was stronger than that at the 3h scale. Meanwhile, for summer $O_3$, due to the quick and strong reactions between precursors, the extracted $O_3$-temperature association at the 3h scale was much stronger than that at the 24h scale. Due to the discrete distribution, the extracted association between all pollutants and precipitation was much weaker at the 3h scale. Similarly, the extracted PM-wind association was notably weaker at the 3h scale. Due to the transport of precursors, summertime $O_3$-wind association was stronger at the 3h scale. For atmospheric pressure, the pollutant-pressure association was weaker at the 3h scale except for summer, when the interactions between atmospheric pressure and other meteorological factors were strong. From the spatial perspective, pollutant-meteorology associations at 3h and 24h were more consistent in those heavily polluted regions, while extracted dominant meteorological factors for pollutants demonstrated more differences at 3h and 24h in those

less polluted regions. This research provides a comprehensive understanding of the effect of temporal scales on pollutant-meteorology association and sheds useful light on better extracting the natural and anthropogenic drivers for airborne pollution.

## Acknowlededgement

This work was supported by the National Natural Science Foundation of China (Grant No.42171399)

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

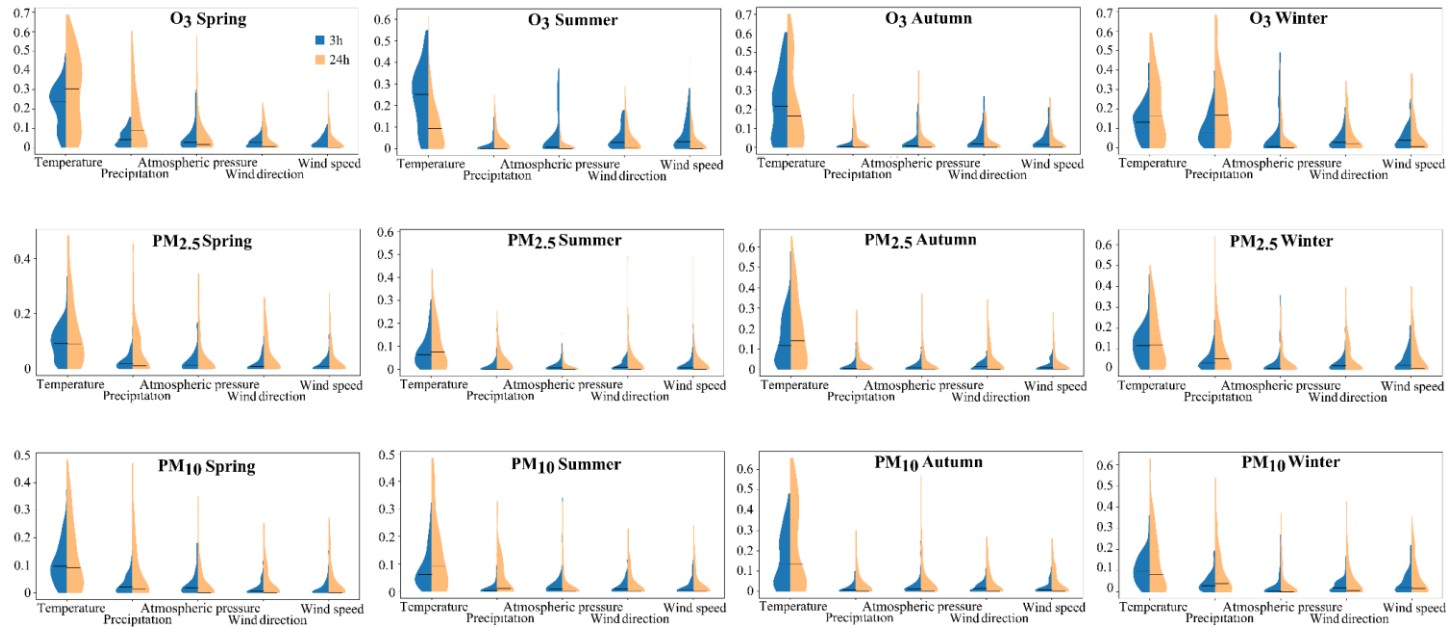

**Figure 1: The violin chart of the ρ value of individual meteorological factors on PM₂.₅, PM₁₀ and O₃ across China at 3h and 24h scale.**

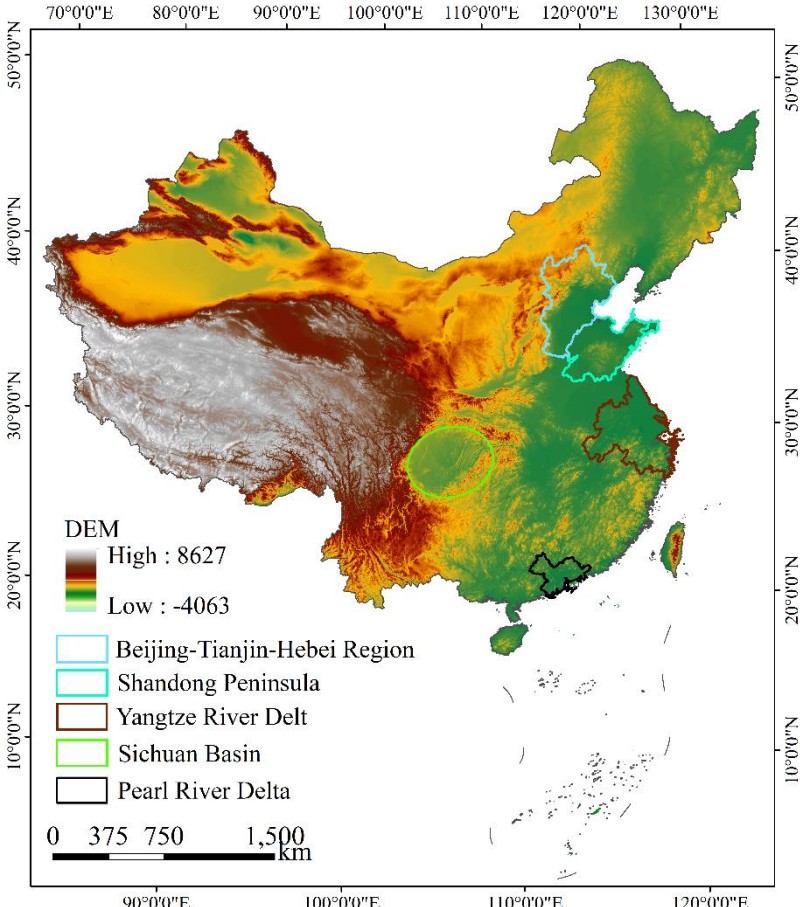

**Figure 2: The Location of all mentioned regions.**

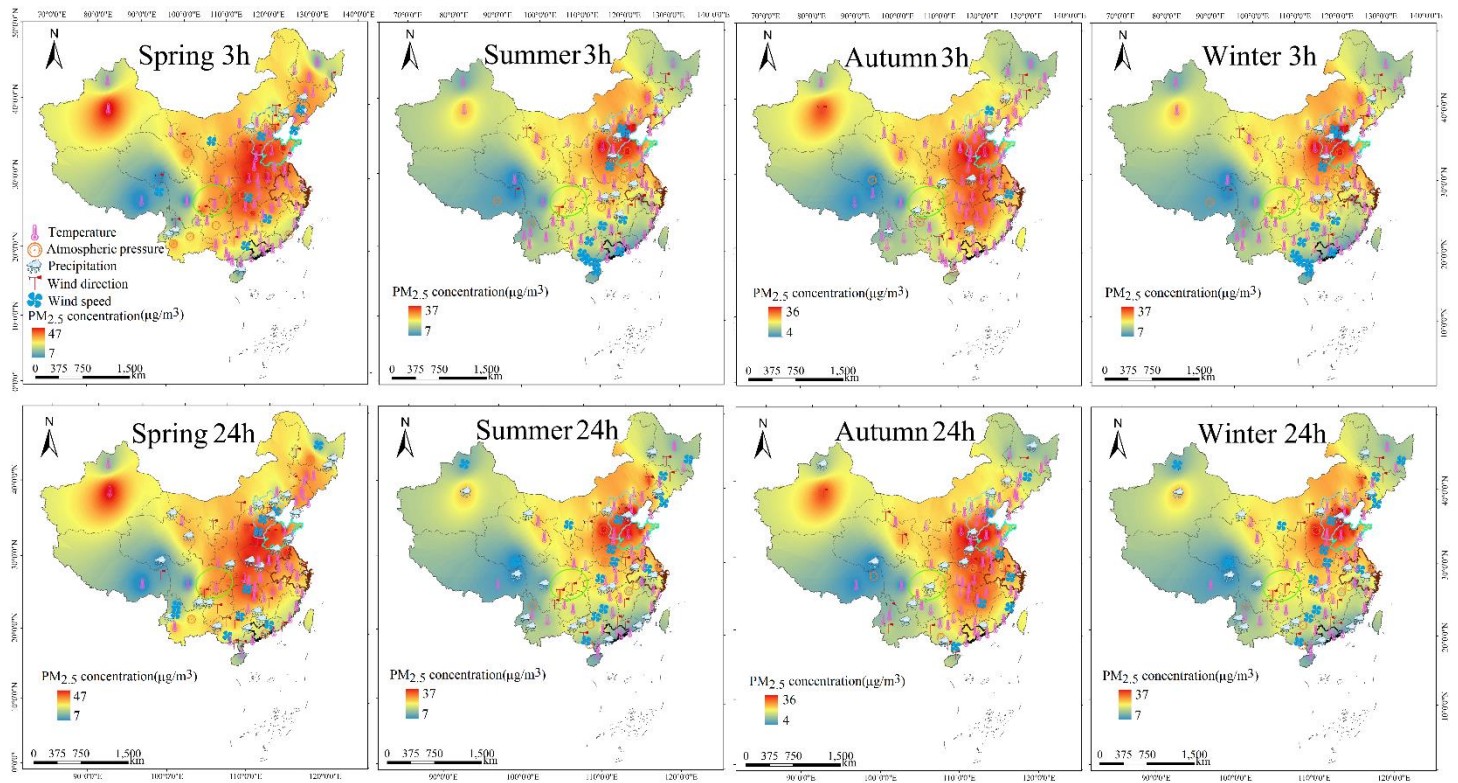

**Figure 3: The dominant meteorological factor for PM2.5 concentrations across China at 3h and 24h scale.**

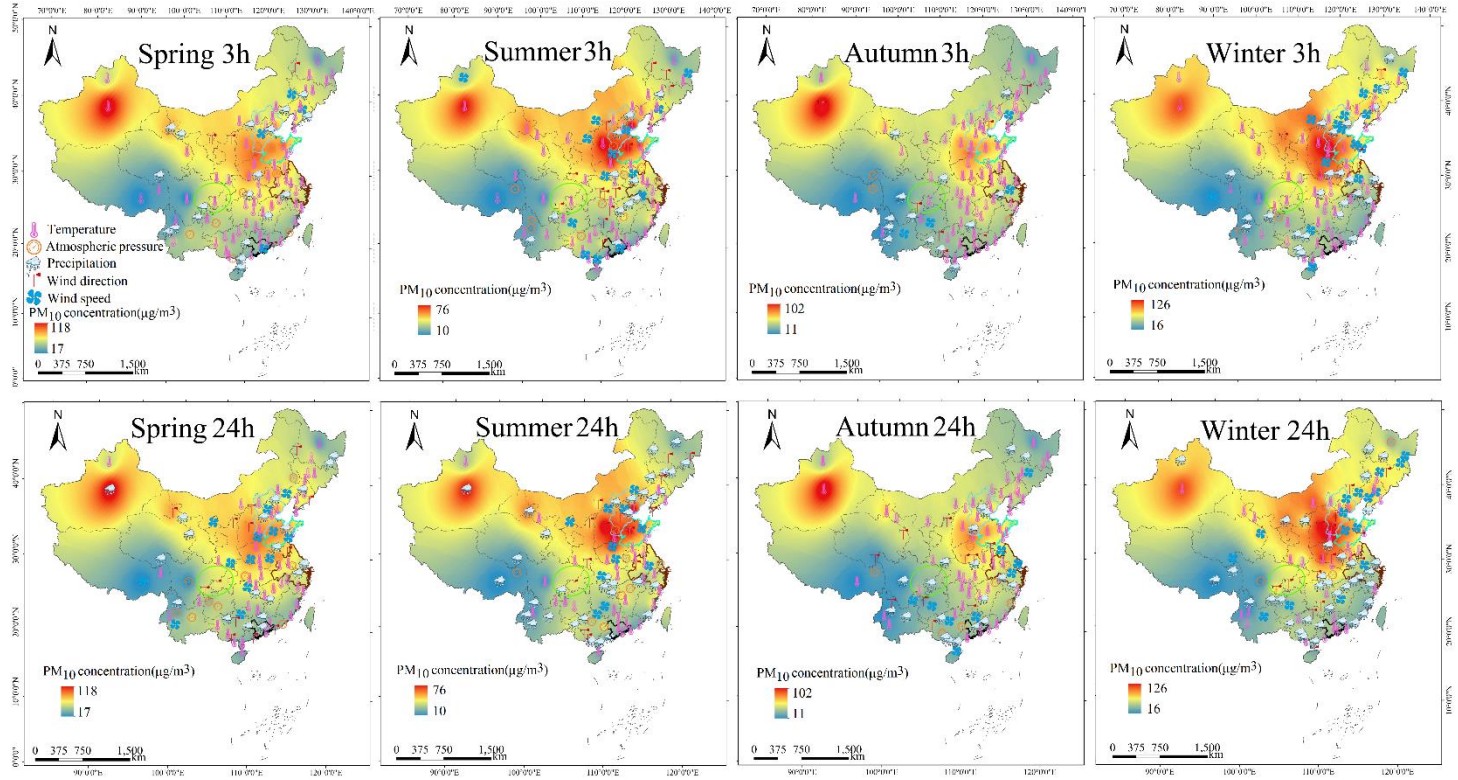

**Figure 4: The dominant meteorological factor for PM$_{10}$ concentrations across China at 3h and 24h scale.**

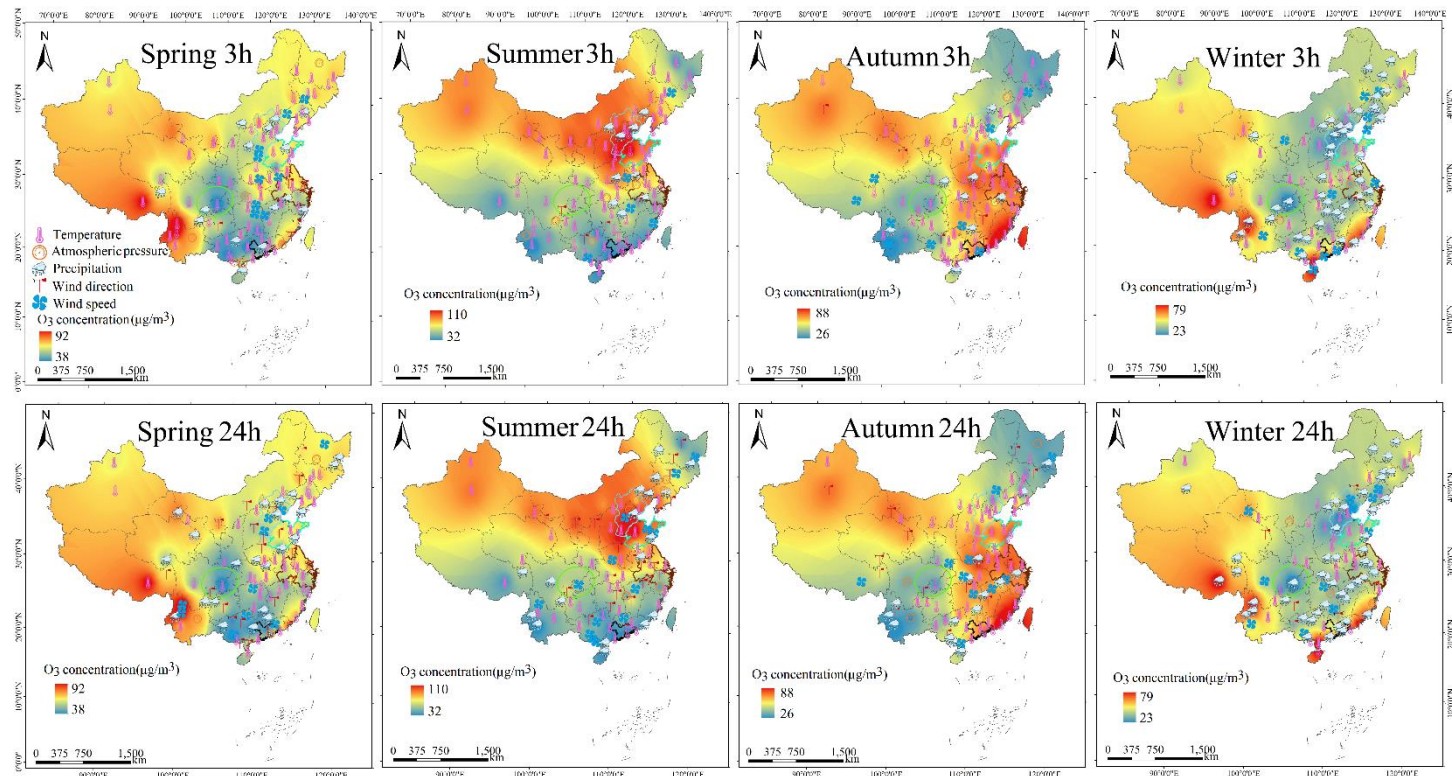

**Figure 5: The dominant meteorological factor for O₃ concentrations across China at 3h and 24h scale.**

**Table 1: The number of cities with this meteorological factor as the dominant meteorological factor for O$_3$.**

| O$_3$ | 3h | | | | 24h | | | |
|---|---|---|---|---|---|---|---|---|
| | Spring | Summer | Autumn | Winter | Spring | Summer | Autumn | Winter |
| Temperature | 64 | 78 | 75 | 42 | 59 | 38 | 58 | 33 |
| Precipitation | 15 | 9 | 8 | 43 | 21 | 18 | 15 | 47 |
| Atmospheric pressure | 7 | 5 | 4 | 3 | 8 | 8 | 4 | 3 |
| Wind Direction | 6 | 4 | 5 | 1 | 8 | 23 | 14 | 7 |
| Wind Speed | 9 | 5 | 9 | 12 | 5 | 14 | 10 | 11 |

**Table 2: The number of cities with this meteorological factor as the dominant meteorological factor for PM$_{2.5}$.**

| PM$_{2.5}$ | 3h | | | | 24h | | | |
|---|---|---|---|---|---|---|---|---|
| | Spring | Summer | Autumn | Winter | Spring | Summer | Autumn | Winter |
| Temperature | 62 | 60 | 79 | 59 | 44 | 43 | 61 | 30 |
| Precipitation | 7 | 9 | 8 | 19 | 22 | 19 | 14 | 35 |
| Atmospheric pressure | 12 | 8 | 4 | 2 | 3 | 8 | 6 | 5 |
| Wind Direction | 12 | 12 | 8 | 6 | 22 | 16 | 13 | 13 |
| Wind Speed | 8 | 12 | 2 | 15 | 10 | 15 | 7 | 18 |

**Table 3: The number of cities with this meteorological factor as the dominant meteorological factor for $PM_{10}$.**

| $PM_{10}$ | 3h | | | | 24h | | | |
|---|---|---|---|---|---|---|---|---|
| | Spring | Summer | Autumn | Winter | Spring | Summer | Autumn | Winter |
| Temperature | 65 | 53 | 73 | 56 | 45 | 34 | 55 | 31 |
| Precipitation | 19 | 11 | 13 | 24 | 20 | 34 | 20 | 36 |
| Atmospheric pressure | 9 | 10 | 4 | 2 | 10 | 8 | 7 | 4 |
| Wind Direction | 4 | 13 | 7 | 4 | 13 | 14 | 10 | 14 |
| Wind Speed | 4 | 14 | 4 | 15 | 13 | 11 | 9 | 16 |

**Table 4: The number of cities with the same dominant factor at both 3h and 24h scale.**

| | spring | summer | autumn | winter |
|---|---|---|---|---|
| $O_3$-meteorological elements | 32 | 42 | 58 | 53 |
| $PM_{2.5}$-meteorological elements | 36 | 42 | 62 | 42 |
| $PM_{10}$-meteorological elements | 42 | 29 | 56 | 43 |

**Table 5: The mean ρ of individual meteorological factors on PM$_{2.5}$, PM$_{10}$ and O$_3$ across China.**

|  |  | Temperature | | Precipitation | | Atmospheric pressure | | Wind Direction | | Wind Speed | |
|---|---|---|---|---|---|---|---|---|---|---|---|
|  |  | 3h | 24h | 3h | 24h | 3h | 24h | 3h | 24h | 3h | 24h |
| O$_3$ | Spring | 0.213 | 0.283 | 0.050 | 0.140 | 0.058 | 0.070 | 0.028 | 0.048 | 0.030 | 0.042 |
|  | Summer | 0.238 | 0.114 | 0.013 | 0.042 | 0.055 | 0.017 | 0.049 | 0.049 | 0.065 | 0.037 |
|  | Autumn | 0.218 | 0.210 | 0.013 | 0.032 | 0.032 | 0.032 | 0.038 | 0.034 | 0.039 | 0.034 |
|  | Winter | 0.133 | 0.198 | 0.100 | 0.191 | 0.058 | 0.035 | 0.045 | 0.058 | 0.052 | 0.062 |
| PM$_{2.5}$ | Spring | 0.095 | 0.128 | 0.027 | 0.059 | 0.030 | 0.034 | 0.018 | 0.048 | 0.015 | 0.030 |
|  | Summer | 0.079 | 0.108 | 0.012 | 0.040 | 0.016 | 0.013 | 0.018 | 0.036 | 0.018 | 0.032 |
|  | Autumn | 0.143 | 0.182 | 0.016 | 0.029 | 0.018 | 0.025 | 0.019 | 0.045 | 0.017 | 0.028 |
|  | Winter | 0.120 | 0.140 | 0.045 | 0.090 | 0.020 | 0.035 | 0.027 | 0.044 | 0.045 | 0.064 |
| PM$_{10}$ | Spring | 0.106 | 0.129 | 0.031 | 0.068 | 0.030 | 0.034 | 0.014 | 0.039 | 0.015 | 0.036 |
|  | Summer | 0.081 | 0.125 | 0.012 | 0.049 | 0.022 | 0.016 | 0.019 | 0.030 | 0.016 | 0.028 |
|  | Autumn | 0.158 | 0.220 | 0.016 | 0.041 | 0.022 | 0.045 | 0.021 | 0.040 | 0.021 | 0.041 |
|  | Winter | 0.109 | 0.127 | 0.046 | 0.082 | 0.020 | 0.029 | 0.028 | 0.050 | 0.043 | 0.063 |