# Peer review of "The role of temporal scales in extracting dominant meteorological drivers of major airborne pollutants"

_Atmospheric Chemistry and Physics, 2022_

## Author Response (AR1)

**Response to reviewers**

Dear Professor Xavier Querol:

**Thanks so much for providing us a chance to revise and resubmit our manuscript. In the past weeks, we have fully revised the manuscript according to all the general and detailed comments raised by two reviewers. The English has been fully polished. Most figures have been reproduced. And the structure and some other issues were also carefully addressed. And for some limitations raised by the reviewers, we have provided detailed explanation in the responses letter and the revised manuscript.**

**Please feel free to contact us if additional revisions are requested and we are more than willing to conduct further revisions.**

**The very best**

**Ziyue**

**To Reviewer 1:**

**Thanks so much for providing us some many general and detailed comments, which helped us so much improve this manuscript. We have fully advised this manuscript according to your constructive comments. We are more than willing to conduct further revisions if you have additional comments.**

**Thanks again for your time and help.**

The article shows an innovative methodology to evaluate the effect of meteorological temporal scales on the pollutant-meteorology association. The collection of a large amount of data on measured pollutant concentrations and meteorological variables makes its results relevant and interesting to address the problem of air pollution in China, more specifically where most of its population is concentrated. The methodology is also interesting for the rest of the scientific community as it is a methodology applicable to other regions.

Although the innovative part of the methodology is clear, it is not clear what is the innovative value of the results obtained. Some of the results seem somewhat obvious or certainly expected, for example, the relationship of temperature with the pollutants analyzed, or a greater association of ozone to the temporal scale of 3h, given its dependence on temperature and solar radiation (in this article not studied as already mentioned by the authors), or the high relationship in winter with precipitation (due to higher frequency of bad weather?). However, the results are no less important for being obvious, but they are shown as if it were a mere descriptive report and there is a lack of greater depth in the analysis of the results and their possible applicability in future strategies for air quality control.

**R: Thanks so much for pointing this out. Yes, we mentioned that temperature was the dominant factor for PM and ozone generally at the national scale, which was clear in our previous studies. However, this was not the major conclusion and findings of this research. As stressed in the manuscript, the major innovation and findings of this research was the temporal scale played an important role in the extracted pollutant-meteorology association. Specifically, despite a generally strong influence of temperature, we found that the number of cities with temperature as the dominant meteorological and the mean ρ value of temperature presented notable differences at the 3h and 24h scale. The difference of pollutant-meteorology association at different temporal scales was not investigated quantitatively before and was the major innovation and contribution of this research.**

**Thanks so much for your comment. In the revised manuscript, we further pointed out that the dominant temperature was clear in previous studies and the temporal effects on revealed difference of pollutant-meteorology association was the new knowledge to the field. In this case, the major innovation of this research was clearly highlighted.**

1. In order to improve this analysis, I propose a series of improvements for the authors to consider:

The reference year studied should appear in the abstract: 2020.

**R: Thanks so much for this comment. We have corrected it accordingly in the revised manuscript.**

2. The structure of the article could be clearer if section 2 were titled Methodology, and 2.2. Advanced Causation Model.

Following the latter, it would be clearer to join section 3 and section 4, since the way the article is presented, it is difficult to follow the relationship between the results and the authors' discussion of them. It would be advisable to make the discussion at the same time as the results are shown.

**R: Thanks so much for this comment. It's a very good suggestion and we have corrected it accordingly in the revised manuscript.**

3. In the introduction, the sentence starting on line 46 "Zhai et al. (2019) estimated... for PM2.5 variations" is not understood.

**R: Thanks so much for this. Here we meant that Zhai et al. (2019) estimated the correlation between $PM_{2.5}$ concentration and meteorological factors at the 10-day scale, found that the variation trend of $PM_{2.5}$ and $SO_2$, $NO_2$ and CO was consistent, also $SO_2$ emission-control was the main driving factor for $PM_{2.5}$ variations. This result was inconsistent with findings from other studies. In the revised manuscript, we have corrected this to make this part clearer.**

4. The introduction should briefly mention why the 3h time scale was chosen and not another. This question is also not addressed in the methodological part of the article. It would also be advisable

to add in the introduction what the purpose of this study is, beyond evaluating the methodology and its results, what future applicability it has, or why this study's results are interesting or innovative.

**R: Thanks so much for pointing this out. This is a good point. Currently, most research concerning pollutant-meteorology association was conducted based on a daily basis (24 hour). This is because meteorology data with a higher temporal resolution is not available. Due to the lack of data sources, there is no research conducted on the temporal effects on the pollutant-meteorology association.**

**We recently obtained the 3h meteorological data sources from China Meteorological Administration, which was so far the meteorological data set with the highest temporal resolution. When hourly data set is not being produced now, we can consider the 3h meteorological data source was the existing data set with the highest temporal scale. Although 3h meteorological source has yet been publically available now, it will become available in the future. In that case, the resolution of 3h and daily (24h) will become the most frequently data set for most researchers. Meanwhile, the comparison between 3h and 24h data set presented a strong difference, indicating the temporal effects on pollutant-meteorology association was strong. With the growing availability of long-term meteorological and pollutant data, multi-scale research is recommended to comprehensively understand the short- and long-term meteorological influences on different airborne pollutants.**

**What you mentioned here was very important. In the revised manuscript, we added relevant explanation.**

5. Regarding the first paragraph of section 2.1. Data sources: Why was the year 2020 selected, is it a question of data availability? Since an annual seasonal study is conducted, how did the emissions change over the year due to the lockdown? This should be stated clearly. Additionally, when analyzing or grouping stations, have any criteria been used to discard any stations?, for example, some with high industrial character or downwind of an industry, or less exposure to the population, or have all of them been taken into account?

**R: Thanks so much for this. As we acknowledged, the 3h meteorological data set was a very rare data set and unfortunately, we did not have a long-term data series with such high temporal-resolution and only got the data set for 2020. This was admitted in the discussion part and we suggested that we would like to explore the multi-year variations of pollutant-meteorology associations. For now, no other studies, to our best knowledge, have conducted analysis using data sets with temporal resolution better than daily. However, the one year data set for this research, including four seasons (four complete time series with more than 90 records (24h) and 720 records (3h)), was sufficient for conducting this comparison analysis. CCM is sensitive to weak to moderate coupling in ecological time series and can effectively extract the association between two variables based on more than 30 records. Therefore, CCM was an ideal tool for conducting this comparison research. Secondly, yes, you are right, the COVID-19 in 2020 may cause may cause a difference in the extracted pollutant-meteorology association. However, by comparing the output of this research with**

our previous studies, the extracted p value was slightly smaller in 2020, yet the general trend was highly consistent with previous studies. Since the major aim of this research was to examine whether temporal scales can cause a difference on extracted pollutant-meteorology association (and it is effectively proved by this research), the specialty of 2020 caused limited influence on the outputs.

**Thanks again for your comment more information on the availability, advantages, and limitations of this data set has been added to the revised manuscript, which helped us improve this manuscript.**

6. How are the location of the 101 cities relevant? Are they the ones shown on the maps in the figures? Do they have any geographic biases that might be relevant to the study, such as different climate zones influencing the results?

**R: Thanks so much for this. The locations of 101 cities are not correlated, they are the ones shown on the maps in the figures. Here we considered all the available cities in this data set, which can demonstrate the overall influence of temporal effects on the extraction of pollutant-meteorology association. Amongst the 101 cities, half of them demonstrated notable difference of dominant meteorological factors between the 3h and 24h scale. So the inclusion of as many as cities across China can effectively remove the potential bias caused by specific regions and highlight the strong temporal effects on pollutant-meteorology association.**

7. It is understood that data used is based on hourly measurements, but they are not mentioned until line 78, is that so?

**R: Thanks so much for this. For this research, the available meteorological data of 3h and 24h were collected respectively. Meanwhile, since the publicly released pollution data was hourly based, to match the temporal scale of meteorological data, the per-3h and per-24h pollutant data were produced by conducting average operation on hourly concentration data.**

8. In section 3.1., moving on to the results, the tables are referenced all at once, they must be referenced one by one as the specific results of each of them are discussed. How relevant are the results shown in this section? The influence in winter is discussed, is winter one of the seasons of concern in terms of pollution problems? And if so, what do these results imply?

**R: Thanks so much for this. This is a very good point. We have corrected it accordingly in the revised manuscript. For all three airborne pollutants, the dominant meteorological factor at the 3h and 24h scale was the same in only a third of cities, indicating the temporal scale played a large role in the analysis of pollutant-meteorology association. From the seasonal scale, the consistence between dominant meteorological factors extracted at 3h and 24h in autumn and winter was higher than that in spring and summer. This phenomenon indirectly suggested that meteorological influences on airborne pollutants were stronger in autumn and winter, and thus the role of dominant meteorological factor was highlighted and notably larger than other factors.**

**Thanks again for this comment. According to this, this part has been largely improved.**

9. In section 3.2. it is mentioned that the relationship between PM and $O_3$ is different, and likewise for the meteorological variables, have the authors identified any dominating meteorological driver that might not be expected or so obvious?

**R: Thanks so much for this. This is a very interesting point. Actually, yes, there seemed some unexpected outputs here. We originally thought that the extracted Pollutant-Meteorology association was always stronger at 3h scale. Interestingly, we found that for some meteorological factors, this was not the same as we expected. For instance, we thought precipitation could exert a stronger influence on PM at 3h scale. However, actually, we found the influence of precipitation on PM was stronger at 24h scale. We assume this was attributed to the characteristics of recording the amount of precipitation at different temporal scales. At the 3h scale, the amount of precipitation at some time slots may be too limited to record, which influenced the calculation outputs.**

10. In section 3.3. the spatial distribution of the dominant meteorological variables is discussed, as in section 3.1., it is necessary to reference each figure and to highlight the relevant results of each one of them.

**R: Thanks so much for pointing this out. We have corrected it accordingly in the revised manuscript.**

11. In line 189 of section 3.3, "heavily polluted summer" is mentioned, does this happen for the whole territory? and is it supposed to be in the areas where a higher concentration of ozone has been plotted? Line 194 talks about "not severe and homogeneous", where readers can observe this?

**R: Thanks so much for this. In summer, ozone pollution is worse than in other seasons. This is because high-temperature and low-humidity are the major driver for ozone pollution, which is often observed in summer. And this scenario is applied to the entire China, as revealed by many previous studies.**

**For PM pollution across China, we generally considered that it was most severe in the three major city-clusters, Beijing-Tianjin-Hebei region, Yangtze-River Delta and Pearl-River Delta. For other regions, especially those inland regions, PM pollution was relatively low and presented less heterogeneous distribution.**

**Thanks so much for your comment. We have revised it accordingly.**

12. Section 3.3. should be accompanied by an indication on the map of "Yangtze River Delt" and "Shandong Peninsula" locations.

**R: Thanks very much for this comment. These figures have been reproduced accordingly.**

13. Regarding section 3.3., the figures (maps) associated with this section are not in the same color scale (for concentrations), nor do they indicate what these concentrations are, the hourly average for each season?

**R: Thanks very much for this comment. Actually, the color bar was set for each season respectively. So the concentration map is only drawn for each season, without overall unification. And these concentrations are the hourly average for each season.**

14. In the first paragraph of Section 4, the limitations of the method are mentioned, it would be more advisable to mention it in the methodology. Lines 211 to 213 reinforce the potential of the study that should have been mentioned previously.

**R: Thanks so much for this. We have corrected it accordingly in the revised manuscript.**

15. The sentence starting on line 225 is not understandable.

**R: Thanks so much for this. Here it means that, Both from a qualitative perspective (the identification of dominant meteorological factors for pollutants) and quantitative perspective (the details of p value of individual meteorological factors on pollutants), the extracted pollutant-meteorology association at 3h and 24h presented large differences, indicating strong temporal effects on pollutant-meteorology associations.**

**Thanks again for your comment. We have revised this part accordingly.**

16. Line 235 talks about "heavily polluted and less polluted seasons", but at no point does it mention which is which.

**R: Thanks so much for this comment. Heavily polluted and less polluted seasons are indicated in the revised version.**

17. The sentence starting on line 237 and ending in 240 should be included into results.

**R: Thanks so much for this. Actually, we have already listed some similar outputs in the result parts "For PM$_{2.5}$ and PM$_{10}$, the calculated influence of temperature at 24h scale was consistently larger than that at the 3h scale. For the relatively cold season winter and spring, when O$_3$ concentrations were relatively low, the influence of temperature at 24h scale was larger than that at the 3h scale. For summer, when O$_3$ concentrations were the highest, the influence of temperature at 3h scale was much larger than that at the 24h scale".**

18. The paragraph starting on line 248 could go to the introduction or part to the method, it does not add value to the discussion of the results.

**R: Thanks so much for this. According to your comment, we have moved this part to the introduction section.**

19. The conclusions in Section 5 are succinct and the results here exposed are clear, although there is no mention of the spatial distribution of the results. This section may be a reference for the authors to improve sections 3 and 4 and to highlight or develop these relevant aspects that are mentioned here.

**R: Thanks so much for this. We have corrected it accordingly in the revised manuscript.**

20. In line 289 "the secondary reaction of which was relatively slow" is not understood.

**R: Thanks so much for this. This means "the secondary reaction between precursors was relatively slow".**

21. Some comments on the wording:

-In "extracted pollutant-meteorlgy", what is meant by "extracted"?

**R: Thanks so much for this. It can also be interpreted as extracted pollutant-meteorology (using CCM).**

The h in 24 is missing: 24h

**R: Corrected. Thanks so much for pointing this out.**

What does the "composite" of "composite airborne pollution" mean?

**R: Thanks so much for this. Composite airborne pollution is a commonly used term, which means the airborne pollution was caused by a variety of atmospheric pollutants.**

Line 87, the authors mean "complex ecosystems"? Instead of "complicated ecosystems"

**R: Corrected. Thanks so much for pointing this out.**

Line 87, what does casual influence mean?, do the authors mean causal influence?

**R: Corrected. Thanks so much for pointing this out.**

Line 92, what does "mirage" correlation mean?

**R: Thanks so much for this. Mirage correlation means the correlation between two variables was not because there are actual causation between them. Instead, there are no causation, and the calculated correlation between them was caused by an agent variable, which was correlated with both of them.  Since CCM was the optimal model to remove the influence of other meteorological factors, the mirage correlation calculated using correlation analysis would be identified by CCM.**

Line 103, there is a typo.

**R: Corrected. Thanks so much for pointing this out.**

The figures are in low resolution, and some symbols are less visible than others, for example the one for temperature.

**R: Thanks so much for pointing this out. The low resolution was caused by the image compression during the uploading process. We have solved it by inserting images with higher resolution.**

**To Reviewer 2:**

Assessing the association between the concentrations of multiple airborne pollutants and driving factors is important for identifying the underlying mechanisms for explaining air pollutants' variations. Xu et al. investigate the effects of temporal scales on the identification of dominant meteorological factors for PM and ozone levels across China in 2020. The results showed that temperature is the most critical meteorological factor at both 3h and 24h scales and pollutant-meteorology associations are in a higher degree of agreement in highly polluted regions.

This work is a good contribution that is meaningful and fills some knowledge gaps to understand the influence of temporal scales in the attribution of airborne pollution in China. I would be glad to see its publication, yet there are still some questions, as elaborated in the article. Please correct and clarify them, which will make this manuscript more reasonable and better present the effects.

**R: Thanks so much for all your constructive remarks and useful suggestions, which has significantly raised the quality of the manuscript. We have addressed the issues you raised in the response letter and the revised manuscript. By clarifying the issues you suggested, the manuscript has been largely improved. Thanks again for all your encouragement and valuable comments.**

**Please feel free to contact us if additional revisions are required and we are more than willing to conduct further revisions according to your comments.**

**Specific comments:**

1. How to extract the reliable association between airborne pollutants and meteorological factors is the key to revealing the temporal efforts on pollutant-meteorology causation, and the selection of robust methods is crucial. Therefore, why CCM is suitable for this research and other models not suitable for such analysis should be clearly explained. In the current form, authors have briefly introduced CCM, yet its principle and advantage remained unclear to me. Please elaborate on the advantages and limitations of CMM model and the advantages and rationality of the CMM model compared with other mainstream models.

**R: Thanks so much for this comment. According to our recent model-comparison paper (Chen et al. 2022), CCM may be the most suitable model for causal inference of atmospheric environment. Theoretically, firstly, CCM is specifically designed to deal with the nonlinear relationship between two variables and is fully suitable for the nonlinear relationship between atmospheric factors. Secondly, CCM automatically considers all possible interaction forms and lag effects between the time series of two variables, which effectively reduces the influence of interference and avoids the influence of other factors. Third, CCM requires less parameter setting and prior knowledge, eliminating the uncertainty caused by improper parameter setting. Therefore, CCM model was an ideal tool for this research.**

**The relevant references and explanation has been added to the revised manuscript. Thanks so much for this comment, which improved the rationality of this research significantly.**

*Chen, Z., Xu, M., Gao, B., et al. Causation inference in complicated atmospheric environment. Environmental Pollution, 2022, 303, 119057.*

2. Some further details in the Discussion section can help explain the motivation and main findings of this study. In particular, some discussions on the related works revealed different dominant meteorological factors when the temporal scale is different. This can re-stress the necessity of considering temporal scales.

**R: As stressed in the manuscript, the major innovation and findings of this research was the temporal scale played an important role in the extracted pollutant-meteorology association. Specifically, despite a generally strong influence of temperature, we found that the number of cities with temperature as the dominant meteorological and the mean ρ value of temperature presented notable differences at the 3h and 24h scale. The difference of pollutant-meteorology association at different temporal scales was not investigated quantitatively before and was the major innovation and contribution of this research.**

**Thanks so much for your comment. In the revised manuscript, we further pointed out that the major meteorological elements of air pollutants was clear in previous studies and the temporal effects on revealed difference of pollutant-meteorology association was the new knowledge to the field. In this case, the major innovation of this research was clearly highlighted.**

**Technical comments:**

1. The time of the data (last access date) should be included according to the requirement of ACP.

**R: Corrected. Thanks so much for pointing this out.**

2. The English is understandable, yet with some typos. I suggest the authors carefully read through the manuscript and correct them. Some examples are listed below:

Line 13: Should be "24h".

Line 95: There's a space missing between "(Chen et al., 2022)" and "proved".

Line 103: There's a missing word.

Line 196: Should be "was".

Line 267: Should be "NO2".

The first letter in " (e.g., ) " is not uppercase or lowercase.

**R: Thanks so much for this comment. We have corrected all these typos in the revised manuscript. Meanwhile, we have re-checked the manuscript carefully and polished the English. Thanks again for your comment.**

**List of all relevant changes made in the manuscript:**

Line 14: 'PM$_{2.5}$, PM$_{10}$ and O$_3$ concentrations at the 3h and 24 scale' → 'PM$_{2.5}$, PM$_{10}$ and O$_3$ concentrations in 2020 at the 3h and 24h scale respectively'.

Line 45: 'between PM2.5 concentration at the 10-day scale and various meteorological factors' → 'between PM2.5 concentration and meteorological factors at the 10-day scale'.

Line 53: 'In recent years, the research on pollutant-meteorology has been massively conducted since 2013 (Chen et al., 2020), yet some gaps remained. Due to the lack of high temporal-resolution data, previous studies were mainly conducted at the daily scale and many scholars may believe that the application of high-temporal-resolution data leads to a better extraction of pollutant-meteorology association.' has been added to the revised manuscript.

Line 62: 'Materials' → 'Methodology'.

Line 81: 'Methods' → 'Advanced Causation Model'.

Line 89: 'casual' → 'causal'.

Line 98: 'CCM is specifically designed to deal with the nonlinear relationship between two variables and is fully suitable for the nonlinear relationship between atmospheric factors.' has been added to the revised manuscript.

Line 104: 'CCM automatically considers all possible interaction forms and lag effects between the time series of two variables, which effectively reduces the influence of interference and avoids the influence of other factors.' has been added to the revised manuscript.

Line 110: 'Based on the rarely employed 3h meteorological data sources, we compared the effects of temporal scales on the extracted pollutant-meteorology causation. Acknowledged, due to the data limitation at the 3h scale, which did not include humidity and sunshine duration, we could simply consider a limited number of meteorological factors (Temperature, Precipitation, Wind Speed, Wind Direction and Atmospheric Pressure), which was less than our previous studies based on meteorological data at the 24h scale, while some meteorological factors (e.g., humility and sunshine duration) were missed in this research. However, since we compared the same set of these major meteorological factors at both 3h and 24h scale, the calculated consistence and difference could effectively reveal the potential effects of different temporal scales on the quantitative (the detailed ρ value) and qualitative (the dominant meteorological factor) findings of pollutant-meteorology association. Despite the limitation of number of meteorological factors, it caused limited influence on the temporal effects on pollutant-meteorology association. This is because CCM simply considers the causality between the target variable and one influencing variable, and removes the influence from other variables (Sugihara et al., 2012; Chen et al., 2020). Another limitation of this data was that this data set simply included one year's data and thus the inter-annual variation of temporal effects on pollutant-meteorology association could not be revealed. For this research, we majorly revealed the existence of strong temporal effects on pollutant-meteorology association, which can be fully supported by the one-year data with four seasons (four complete time series with more than 90 records (24h scale) and 720 records (3h scale)Meanwhile,

the temporal variation of temporal effects on pollutant-meteorology association and its influencing factors should be further investigated in future studies, when the long time series data sets of 3h meteorological data become available.' has been added to the revised manuscript.

Line 132: ' (As shown in Table 1, Table 2 and Table 3) ' → 'O$_3$ (Table 1), PM$_{2.5}$ (Table 2) and PM$_{10}$ (Table 3)'.

Line 150: 'which was consistent with previous studies (Wang et al., 2018; Yang et al., 2021)' has been added to the revised manuscript.

Line 211: '(Figure 3)', '(Figure 4)' has been added to the revised manuscript.

Line 215: '(Figure 5)' has been added to the revised manuscript.

Line 221: 'are' → 'was'.

Line 232: 'Based on the …' has been removed.

Line 261: '(e.g. The heavily polluted season for O$_3$ and PM was winter and summer respectively).' has been added to the revised manuscript.

Line 274: 'In recent years …' has been removed.

Line 290: '(e.g., humidity, boundary layer height)' → '(e.g., Humidity, Boundary layer height)'.

Line 231: 'the influence of meteorological factors on airborne pollutants has obvious seasonal variations and presented some regional similarity,' has been added to the revised manuscript.

---

## Author Response (AR2)

**Response to Editor and reviewers**

**Dear Professor Xavier Querol:**

**Thanks so much for providing us a chance to revise and resubmit our manuscript, when one reviewer clearly misunderstood the key innovation and contribution of this research. As supported by the other three reviewers during the first and second round of review process, this research provided new visions for the communities on the strong temporal effects on pollutant-meteorology associations across China, which has rarely been investigated before due to the lack of high-resolution temporal data. The other three reviewers all agreed that this research was novel and presented some interesting findings. Specifically, when previous scholars suggested that pollutant-meteorology associations could present large variations at different temporal scales, yet no quantitative research has been conducted to prove or further interpret this issue. This research revealed that the temporal effects might be even stronger than expected, characterized with more than 50% of cities presented different dominant meteorological factor for specific pollutant at 3h and 24h scale. We also noticed that the role of some meteorological factors, such as wind and precipitation, may be largely underestimated at the 3h scale. This research also found that the temporal effects were stronger in those less polluted regions. As a team which has published many papers concerning pollutant-meteorology associations, we guarantee these findings are new and have yet been discussed by us or other scholars before. All these conclusions suggest that we should be careful to attribution airborne pollution from both the anthropogenic and meteorological perspectives by choosing temporal scale carefully and better employing the combination of multiple-scale data.**

**The reviewer mentioned that some content was similar to previous studies. I think this is a major misunderstanding. Actually, this type of content only accounted for a very small proportion and simply served as the introduction of related studies. As you can see in the similarity report, the similarity between this research and previous studies was very low, indicating the findings from this research are quite different from previous studies and we mainly focus on the difference of pollutant-meteorology associations at 3h and 24h (the**

temporal effects), instead of pollutant-meteorology associations at one specific time scale. As one can see in the abstract and conclusion part, all the major findings from this research were completely new and none part of that was revealed before.

Although the lack of new findings and the confusion of the major purpose of this research is purely a misunderstanding, we are still willing to make substantial efforts to address this confusions. We have added more references to the introduction and discussion part to further emphasize the importance of revealing the temporal effects on pollutant-meteorology associations, which can highlight the innovation and major contributions of this research. We have further added much more content to the results part, especially the spatial patterns of temporal effects on pollutant-meteorology associations. In total, we added around more than 800 words to further illustrate our research. In this case, we believe the purpose and value of this research can be better understood.

Thanks again for processing our manuscript and keeping giving us chances to improve it. I believe that as an top scholar and editor in the field of atmospheric pollution research, you can fully understand the value of this research after you checked the revised version and above explanation.

Please feel free to contact us if additional revisions are requested and we are more than willing to conduct further revisions.

The very best

Ziyue

**To Reviewer 1:**

**Thanks so much for providing us some many general and detailed comments, which helped us so much improve this manuscript. We have advised this manuscript according to your constructive comments. We are more than willing to conduct further revisions if you have additional comments.**

**Thanks again for your time and help.**

It is clear that the methodology is innovative, but despite the exhaustive review made previously, the purpose of this methodology is not expressed or detailed, nor what applications it may have. In fact, many of the results shown are obvious and consistent with what is already known by the scientific community, as for example paragraph starting in line 147 and lines starting in 167 and 319 (pressure associated discussion). Additionally, figures have not been improved and neither are self-explanatory.

**R: Thanks so much for your comments. Actually, I think there is a misunderstanding and we are sorry that we did not make it clearer here. The use of CCM, which has been frequently employed by us and other scholars, is not innovative and the major contribution of this research. Instead, as supported by the other three reviewers during the first and second round of review process, the major highlight and contribution of this research is to provide new visions for the communities on the strong temporal effects on pollutant-meteorology associations across China, which has rarely been investigated before due to the lack of high-resolution temporal data. Specifically, when previous scholars suggested that pollutant-meteorology associations could present large variations at different temporal scales, yet no quantitative research has been conducted to prove or further interpret this issue. Based on the rare national 3h meteorological data sources (the major highlight and innovation of this research, which to our best knowledge, the highest temporal-resolution national data sources ever employed), this research revealed that the temporal effects might be even stronger than**

expected, characterized with more than 50% of cities presented different dominant meteorological factor for specific pollutant at 3h and 24h scale. We also noticed that the role of some meteorological factors, such as wind and precipitation, may be largely underestimated at the 3h scale. This research also found that the temporal effects were stronger in those less polluted regions. As a team which has published many papers concerning pollutant-meteorology associations, we guarantee these findings are new and have yet been discussed by us or other scholars before. All these conclusions suggest that we should be careful to attribution airborne pollution from both the anthropogenic and meteorological perspectives by choosing temporal scale carefully and better employing the combination of multiple-scale data. And we believe this is the key purpose, major findings and interpretation (applications) of this research. These introduction, findings and discussion are all made on the temporal effects on pollutant-meteorology associations across China, instead of the causation model CCM.

You mentioned that some content was similar to previous studies. Actually, this type of content only accounted for a very small proportion and simply served as the introduction of related studies. As you can see in the similarity report, the similarity between this research and previous studies was very low, indicating findings from this research are quite different from previous studies and we mainly focus on the difference of pollutant-meteorology associations at 3h and 24h (the temporal effects), instead of pollutant-meteorology associations at one specific time scale. Furthermore, you can see in the abstract and conclusion part, all the major findings from this research were completely new, which are all about the difference between 24h and 3h, and none part of that was revealed before.

Thanks again for pointing out you concerns. To address the potential confusions, we are still willing to make substantial efforts to address this confusions. We have added more references to the introduction and discussion part to further emphasize the importance of revealing the temporal effects on pollutant-meteorology associations, which can highlight the innovation and major contributions of this research. We have further added much more content to the results part, especially the spatial patterns of temporal effects on pollutant-meteorology associations. In total, we added around more than 800 words to further

illustrate our research. In this case, we believe the purpose and value of this research can be better understood.

Meanwhile, according to your suggestion, the locations of "Yangtze River Delta" and "Shandong Peninsula" have been indicated on the map. The legend is indicated in the second small image in Figure 2, 3, and 4. The Yangtze River Delta region has a green border, while the Shandong Peninsula has a blue border. Enlarged figure s can provide a clearer view.

Thanks again for your continuous help during two rounds of review. Please feel free to contact us if additional revisions are requested and we are more than willing to conduct further revisions.

**List of all relevant changes made in the manuscript:**

Line 23: 'From the spatial perspective, pollutant-meteorology associations at 3h and 24h were more consistent in those heavily polluted regions, while extracted dominant meteorological factors for pollutants demonstrated more difference at 3h and 24h in those less polluted regions.' has been added to the revised manuscript.

Line 50: 'Fu et al. (2020) used integrated empirical mode decomposition (EEMD) to decompose the time series data of $PM_{2.5}$, five other atmospheric pollutants and six meteorological types. On the daily scale, $PM_{2.5}$ was positively correlated with $O_3$ and daily maximum and minimum temperature, and negatively correlated with air pressure, while $PM_{2.5}$ presented an opposite association with these factors at the monthly scale.

Despite massive studies conducted, notable inconsistence of dominant meteorological and anthropogenic drivers for airborne pollutants was observed between findings from previous studies. Even if some studies revealed different pollutant-meteorology association at multiple temporal scales, such research conducted in isolated cities, cannot reflect the spatiotemporal variations of temporal effects across China.' has been added to the revised manuscript.

Line 62: 'Due to' → 'More importantly, due to'.

Line 63: 'at the daily scale and many scholars' → 'at the daily scale, while many scholars'.

Line 120: 'We obtained the 3h meteorological data sources from China Meteorological Administration.' has been added to the revised manuscript.

Line 160: 'summer and autumn' has been removed.

Line 188: 'High temperature promotes photochemical reactions and produce more $PM_{2.5}$, $PM_{10}$ and other precursors of secondary pollutants, leading to higher concentrations of $PM_{2.5}$ and $PM_{10}$. High temperature may also lead to increased evaporation loss of $PM_{2.5}$ and $PM_{10}$, including $NO^{3-}$ salt and other volatile or semi-volatile components, resulting in decreased concentrations of $PM_{2.5}$ and $PM_{10}$.' has been added to the revised manuscript.

Line 206: 'at 3h scale' → 'at 3h scale in summer'.

Line 213: 'This may be attributed to existence of the Asian monsoon system, which includes the strong southeast and southwest summer monsoon in China.' been added to the revised manuscript.

Line 234: 'Based on the extracted pollutant-meteorology associations at 3h scale, which have rarely been discussed, we found some interesting differences of pollutant-meteorology association between 3h and 24h in some major regions across China. For the heavily polluted Beijing-Tianjin-Hebei Region, the dominant meteorological factor for $O_3$ in spring was temperature at 3h scale. Meanwhile, the dominant factor was wind speed at the 24h scale. For $PM_{2.5}$, the dominant factor for $PM_{2.5}$ in spring was temperature at the 3h scale and wind speed at the 24h scale. The dominant meteorological factor for $PM_{10}$ in summer was temperature at the 3h scale and precipitation at 24h scale.

For the Yangtze River Delta, the dominant meteorological factor for $O_3$ in spring was temperature at the 3h scale and the combination of temperature and precipitation at the 24h scale. In summer, the dominant meteorological factor for $O_3$ was temperature at the 3h scale and wind speed at the 24h scale. The dominant factor of $PM_{2.5}$ in spring was temperature at the 3h scale and the combination of temperature and precipitation at the 24h scale. The dominant factor of $PM_{10}$ in spring was mainly temperature at the 3h scale and wind speed at the 24h scale. For the Pearl River Delta, the dominant meteorological factor for $O_3$ in winter was temperature at the 3h and precipitation at the 24h scale.

For Sichuan Basin, the dominant meteorological factor for $O_3$ in all four seasons was constantly temperature at the 3h time scale, while it was precipitation, atmospheric pressure and wind speed in summer, autumn and winter respectively at the 24h scale. The dominant meteorological element for $PM_{2.5}$ was temperature in all four seasons at the 3h scale, while it was precipitation in summer and winter at the 24h scale. The dominant meteorological element for $PM_{10}$ in spring and winter was temperature at the 3h scale, while it was atmospheric pressure for spring and winter at the 24h scale. Compared with other regions, the unique basin terrain led to stronger temporal effects on extracted pollutant-meteorology associations.

Our previous (Chen et al., 2018; 2020) revealed that meteorological influences exerted a stronger influence on PM pollutants when PM concentration was higher. This might be the reason that the difference of PM-meteorology associations between 3h and 24h was relatively small in heavily polluted winter and large in less-polluted spring. Meanwhile, we found that the role of wind speed and precipitation may be largely underestimated at the 3h scale. Compared with the generally consistent pollutant-meteorology associations in these heavily polluted regions, the dominant factor for PM and $O_3$ demonstrated significant variations in those coastal cities, such as Shenzhen, Zhuhai, Zhanjiang.' has been added to the revised manuscript.

Line 298: 'pointed out the difference' → 'pointed out the notable differences'.

Line 299: 'few studies have actually conducted the quantitative analysis due to the lack of data.' → 'and the great importance to better understand the temporal effects, few studies actually conducted a comparative analysis due to the lack of data, especially the high temporal resolution meteorological data.'

Line 307: 'we got several interesting and useful findings as follows' → 'we got some major conclusions as follows'.

Line 309: 'difference' → 'differences'.

Line 313: 'The role of wind speed and precipitation, which may be recognized as dominant meteorological factors at the 24h scale, can be largely underestimated at 3h scale.' has been added to the revised manuscript.

Line 371: 'From the spatial perspective, pollutant-meteorology associations at 3h and 24h were more consistent in those heavily polluted regions, while extracted dominant meteorological factors for pollutants demonstrated more differences at 3h and 24h in those less polluted regions.' has been added to the revised manuscript.

Line 373: 'for the' → 'of the'.

Line 374: 'scale' → 'scales'.

Line 376: '**Acknowledegement** This work was supported by the National Natural Science Foundation of China (Grant No.42171399)' has been added to the revised manuscript.

Line 402: 'Fu, H., et al. 2020. Investigating PM2. 5 responses to other air pollutants and meteorological factors across multiple temporal scales. Scientific reports, 10(1), 1-10.' has been added to the revised manuscript.

---

## Author Response (AR3)

**Response to Editor and reviewers**

**Dear Professor Xavier Querol:**

**Thanks so much for providing us a chance to revise and resubmit our manuscript. In the past weeks, we have fully revised the manuscript according to all the general and detailed comments raised by the reviewer.**

**Please feel free to contact us if additional revisions are requested and we are more than willing to conduct further revisions.**

**The very best**

**Ziyue**

**To Reviewer 1:**

The document has been significantly improved, and considering the authors' emphasis on methodology, I have carefully reviewed the manuscript. Changes should be made, especially in how the results of sections 3.1 and 3.3 are presented. As a reader, there are instances where the described results are not found, making it difficult to locate or contradictory to what is stated. I also suggest making minor changes.

**R: Thanks so much for your encouragement and providing us another chance to further improve this manuscript. We have advised this manuscript fully according to your constructive comments. We are more than willing to conduct further revisions if you have additional comments.**

**Thanks again for your time and help.**

Issues regarding results and their comprehension:

-In section 3.1.

• It would be helpful to add the percentage of cities in Table 4 (if applicable) when referring to "a third of cities" in line 139. Additionally, indicating the total number of cities again in the same table or adding percentages to all the cities listed would be beneficial.

**R: Thanks so much for this comment. The city percentage has been added to Table 4. Specific percentages have also been added in the corresponding sections of the text.**

• Regarding line 140, "From the seasonal scale... was higher than that in spring in summer," it is advisable to provide a specific number from the tables or reference the table once again. Alternatively, consider highlighting the relevant numbers in the table. Currently, when referring to the tables, it becomes challenging for the reader to understand which specific table is being referenced.

**R: Thanks so much for this comment. It's a very good suggestion and we have corrected it accordingly in the revised manuscript.**

• Throughout the text, no numbers from the tables are mentioned. It is recommended to include these numbers in the text to help the reader follow the authors' indications.

**R: Thanks so much for this comment. It's a very good suggestion and we have corrected it accordingly in the revised manuscript.**

• The paragraph in line 151, "at the 3h scale, for O3..." refers to Table 1. It is necessary to re-reference Table 1 in this context. It would be helpful to highlight the relevant information in the table, using bold or shading. Similarly, when mentioning $PM_{2.5}$ and $PM_{10}$, which are in the other table, I suggest employing the same approach.

**R: Thanks so much for this comment. It's a very good suggestion and we have corrected it accordingly in the revised manuscript.**

"As a comparison, at the 24h scale, for O3, PM2.5, and PM10, the number of cities with temperature as the dominant influencing factor was largest in spring, which was consistent with previous studies (Wang et al., 2018; Yang et al., 2021), while the number of cities with precipitation was largest in winter." For ozone, I see 59 cities in spring compared to 58 in autumn. For PM2.5, there are 61 cities in autumn versus 44 in spring. Hence, spring seems to be the dominant season for PM2.5?. Regarding PM10, there are 36 cities in winter compared to 34 in summer, indicating a similar distribution between the seasons.

**R: Thanks so much for your comment. Yes, as you pointed here, we did not clearly and correctly explained the pollutant-meteorology association in different seasons. We have fully revised this part according to your comment.**

**Again, thanks so much for pointing this out.**

• The sentence starting at line 159, "For both the 3h and 24h... majority of cities," lacks specific information. I suggest removing this sentence.

**R: Thanks so much for pointing this out. This sentence has been deleted.**

• The following sentence, "the consistency of dominant factors between two temporal scales remained less than 50%," is unclear. It is unclear what it is referring to or if there is a specific data point in the table.

**R: Thanks so much for this comment. Actually, when the extracted dominant factor at the 3h and 24h scale was the same, then we say it was consistent. Then as shown in Table 4, the number of cities with the same meteorological dominant factors at both 3h and 24h was less than 50% of the total number of cities. Therefore, we mentioned the consistency was less than 50%.**

• The sentence that follows, "This may be attributed... could not be revealed," is difficult to understand. It is unclear what is being conveyed in this statement.

**R: Thanks so much for this comment. The study identified the dominant meteorological factors through CCM according to the ρ value. While ρ of the dominant meteorological factor was largest, it may be just slightly larger than ρ of other meteorological factors at 24h (3h) scale, and may be smaller than ρ of another factor, which led to the change of dominant factor, at 3h (24h) scale. In this case, if we simply consider the difference between qualitative output ( just the dominant meteorological factor with the largest ρ) revealed at 3h and 24h scale to reveal the temporal effects of pollutant-meteorology association, the analysis was not complete. Therefore, we also presented the detailed ρ value between all available meteorological factors and these three pollutants acquired 3h and 24h scale.**

**Thanks so much for your comment. We have corrected**

-In section 3.2.

• The statement in line 175, "indicating that meteorological influences on particulate matters and gaseous pollutants were different," is evident and already known. It should be highlighted only if the authors have discovered some valuable additional insights or if it is the first time they confirm it for a specific case.

**R: Thanks so much for this.**

• When discussing the influence of temperature on O3, it would be relevant to mention any previous findings that also identify a connection with wind (which is later mentioned in the spatial distribution and conclusions) when referring to O3 transport. Could these areas be downwind of major emission sources?

**R: Thanks so much for this. Regarding factors that were not considered in the study regarding these areas and emission sources, the study directly used observation data from atmospheric pollutant monitoring stations.**

• If the seasonal variations in precipitation intensity are evident in the table, it should be explicitly mentioned in the text.

**R: Thanks so much for this comment. It's a very good suggestion and we have corrected it accordingly in the revised manuscript.**

• The sentence "This may be attributed to the existence of the Asian monsoon system, which includes the strong southeast and southwest summer monsoon in China" should only be included if there are references to dispersion studies that confirm this claim. It is important to provide supporting evidence for such statements.

**R: Thank you for pointing out this point. It has been revised accordingly.**

-In section 3.3.

• The order of the figures should be adjusted so that PM2.5 is Figure 2, as it is currently mentioned first.

**R: Thanks so much for pointing this out. We have corrected it accordingly in the revised manuscript.**

• Since various regions of China are discussed, it would be helpful to add a map indicating these regions in addition to the existing maps. The regions mentioned, such as the Shandong Peninsula, Beijing-Tianjin-Hebei, Yangtze River Delta, Sichain Basin, Shenzhen, etc., can be marked with circles or another legend on the maps. The added legend to the maps is not clear at all.

**R: Thanks so much for this. This is a good suggestion. Accordingly, we have added a location map to the revised manuscript to demonstrate the locations of all mentioned regions. Meanwhile, we use unified borderlines with different colors to demonstrate these regions in**

**relevant maps. Thanks again for this valuable comment, which make the findings of this research much clearer.**

• In Figures 2, 3, and 4, it is important to provide a detailed description of what the concentration represents, at least in the figure caption. This could include specifying that it is the average of the data used.

**R: Thanks so much for this comment. It really helps. The required information has been added to these figures.**

• I still recommend using consistent color scales across the figures for each pollutant. Currently, there is a difference of only 1-2 micrograms in the upper and lower ranges of some figures. It would be more effective to use a scale that is consistent with the specific pollution issue being addressed and the relevant control metrics associated with it.

**R: Thanks so much for pointing this out. Yes, we fully agree with you that a consistent color bar would be better. However, as you noticed that, in some figures, the range was simply 1-2 micrograms. We have tried the mapping strategy you suggested. If we use a unified color bar, then the 1-2 micrograms presented nearly no difference and thus the entire background would seems a plain color without any differences. In this case, we have to use different color bars for different seasons and pollutants. Actually, this issue occurred before, and thus we had to use different color bars in many of our previous publications (e.g.)**

*Chen, Z., Chen, D., Zhao, C., et al., 2020. Influence of meteorological conditions on $PM_{2.5}$ concentrations across China: A review of methodology and mechanism. Environment International 139:105558.*

*Chen, Z., Li, R., Chen, D., et al. 2020. Understanding the causal influence of major meteorological factors on ground ozone concentrations across China. Journal of Cleaner Production. 242, 118498.*

*Chen, Z., Xie,X., Cai, J. et al, 2018. Understanding meteorological influences on PM$_{2.5}$ concentrations across China: a temporal and spatial perspective, Atmospheric Chemistry and Physics. 2018, 18(8):5343-5358)*

**So again, thanks so much for pointing out this valuable comment and we fully understand and agree with you. And please understand that it may cause some mapping issues if the unified color bar is employed to different maps.**

In section 4, after "...variation of specific meteorological factors (e.g., Temperature) exerted a stronger influence on PM and O3 than the daily variation," the authors should include a comment related to pollutant transport since it is mentioned in the conclusions (line 333). This would help provide a more comprehensive understanding of the findings and their implications.

**R: Thank you for pointing out this point. It has been revised and improved.**

Minor suggestions:

-The term "complicated exosystems" appears again when I believe the authors intended to use "complex" in other contexts (line 94). Please confirm if this is the case.

**R: Yes, this is a very good comment. We have revised all "complicated" to "complex" in the revised manuscript according to your comment.**

-In the last sentence of the introduction (line 66), it seems to be missing some words such as "emission-cut policies," for example. Please provide the specific addition or clarification needed.

**R: Thanks so much for pointing this out. We have corrected it accordingly in the revised manuscript.**

-In line 112, when enumerating t, E, and b, it would be advisable to maintain the same order as previously described.

**R: Thanks so much for pointing this out. We have corrected it accordingly in the revised manuscript.**

-The beginning of the third paragraph in section 2 (line 135), "We obtained the 3h meteorological data sources from China Meteorological Administration," should be moved to section 2.1, which is the Data Sources subsection.

**R: Corrected. Thanks so much for pointing this out.**

-In line 306, "On the other hand?"

**R: Corrected. Thanks so much for pointing this out.**

-In line 323, it should be "Results..."

**R: Corrected. Thanks so much for pointing this out.**

-In line 329, it is recommended to rephrase "the secondary reaction of which was relatively low" as it sounds awkward in English. Please provide an alternative phrasing or clarify the intended meaning.

**R: Thanks so much for this comment. It's a very good suggestion and we have corrected it accordingly in the revised manuscript.**

**List of all relevant changes made in the manuscript:**

Line 66: 'emission-cut' → 'emission-cut measures'.

Line 90: 'complicated' → 'complex'.

Line 94: 'complicated' → 'complex'.

Line 99: 'complicated' → 'complex'.

Line 103: 'complicated' → 'complex'.

Line 115: 'We obtained the 3h meteorological data sources from China Meteorological Administration.' has been removed.

Line 139: 'a third of cities' → '31.68% ~ 61.29%'.

Line 140: 'From' → 'As can be seen from Table 1, Table 2 and Table 3, from'.

Line 142: 'For example, temperature, precipitation, etc., $O_3$, $PM_{2.5}$, and $PM_{10}$ were mostly more dominant in autumn and winter than in spring and summer.' has been added to the revised manuscript.

Line 153: 'At' → 'As can be seen from Table 1, at'.

Line 154: 'with 43 cities,' has been added to the revised manuscript.

Line 155: '; For' → ', with 64 cities, 78 cities, and 75 cities, respectively; As can be seen from Table 2 and Table 3, for'.

Line 156: '$O_3$, $PM_{2.5}$ and $PM_{10}$,' → ', $O_3$,'.

Line 157: 'with 59 cities, and for $PM_{2.5}$ and $PM_{10}$, the number of cities with temperature as the dominant influencing factor was largest in autumn, with 61, 55 cities, respectively,' has been added to the revised manuscript.

Line 160: ', with 47, 35, and 36 cities, respectively' has been added to the revised manuscript.

Line 164: 'For both the 3h and 24h scale, we could see temperature and precipitation exerted strong influences on $O_3$, $PM_{2.5}$ and $PM_{10}$ in the majority of cities.' has been removed.

Line 165: 'The study identified the dominant meteorological factors through CCM according to the $\rho$ value. While $\rho$ of the dominant meteorological factor was largest, it may be just slightly larger than $\rho$ of other meteorological factors at 24h (3h) scale, and may be smaller than $\rho$ of another factor, which led to the change of dominant factor, at 3h (24h) scale. In this case, if we simply consider the difference between qualitative output (just the dominant meteorological factor with the largest $\rho$) revealed at 3h and 24h scale to reveal the temporal effects of pollutant-meteorology association, the analysis was not complete.' has been added to the revised manuscript.

Line 170: 'This may be attributed to the fact that the extraction of dominant meteorological factor amongst several factors was relatively qualitative and thus some subtle differences between different meteorological factors could not be revealed.' has been removed.

Line 210: 'The eastern region of China is affected by summer monsoon in summer and autumn, there is a lot of precipitation; In winter, China receives less precipitation due to the influence of winter winds.' has been added to the revised manuscript.

Line 220: 'This may be attributed to existence of the Asian monsoon system, which includes the strong southeast and southwest summer monsoon in China.' has been removed.

Line 229: 'As shown in Figure 2, all the locations of the mentioned regions have been marked.' has been added to the revised manuscript.

Line 230: 'The seasonal concentration of air pollutant data for each city is calculated using the average of hourly concentration data measured by all available local observation stations.' has been added to the revised manuscript.

Line 236: '$O_3$ (Figure 2)' → '$O_3$ (Figure 5)'.

Line 278: 'Figure 5 inserted here.' has been added to the revised manuscript.

Line 298: '(e.g. Temperature)' → '(e.g. Temperature, Wind speed)'.

Line 299: 'The concentrations of PM and $O_3$ largely depend on wind conditions. High $O_3$ concentrations in different cities usually occur in the presence of strong wind speed, but are independent of wind direction, while high PM is often accompanied by weak wind speed, poor dispersion conditions, and sometimes occurs in strong northerly or southerly winds. The regional transport of air pollutants between cities is common (Li et al., 2019).' has been added to the revised manuscript.

Line 312: 'complicated' → 'complex'.

Line 319: 'complicated' → 'complex'.

Line 326: 'On the other' → 'On the other hand'.

Line 343: 'The result' → 'Results'.

Line 349: 'the secondary reaction of which was relatively slow' → 'the secondary reaction was relatively mild'.

Line 418: 'Li, X., Hu, X., Shi, S., Shen, L., Luan, L., Ma, Y.: Spatiotemporal variations and regional transport of air pollutants in two urban agglomerations in northeast china plain, Chin. Geogr. Sci. 29, 917–933, https://doi.org/10.1007/s11769-019-1081-8, 2019.' has been added to the revised manuscript.

Line 470: Figure 2 has been revised.

Line 474: 'Figure 2: The dominant meteorological factor for O3 concentrations across China at 3h and 24h scale.' → 'Figure 2: The dominant meteorological factor for PM2.5 concentrations across China at 3h and 24h scale.'.

Line 476: Figure 3 has been revised.

Line 479: Figure 4 has been revised.

Line 482: Figure 5 has been added.

Line 483: 'Figure 5: The dominant meteorological factor for O3 concentrations across China at 3h and 24h scale.' has been added to the revised manuscript.

---

## Author Response (AR4)

**Response to Editor and reviewers**

**Dear Professor Xavier Querol:**

**Thanks so much for providing us a chance to revise and resubmit our manuscript. We have fully revised the manuscript according to all the comments raised by the reviewer.**

**Please feel free to contact us if additional revisions are requested.**

**The very best**

**Ziyue**

**To Reviewer 1:**

**R: Thanks so much for your encouragement and providing us another chance to further improve this manuscript. We have advised this manuscript fully according to your constructive comments. We are more than willing to conduct further revisions if you have additional comments.**

**Thanks again for your time and help.**

I would like to reflect only a few minor comments and recommendations:

-Line 69: Replace "per 3hour" with "3 hourly."

**R: Thanks so much for this comment. It's a very good suggestion and we have corrected it accordingly in the revised manuscript.**

-Line 138: Make it clear what each percentage corresponds to; it's not clear. I understand that one is for 3 hours and the other for 24 hours?

**R: Thanks so much for this comment. Actually, as show in Table 4, the consistence between dominant meteorological factors for PM$_{2.5}$, PM$_{10}$ and O$_3$ at two temporal scales varied significantly, ranging from 31.68% ~ 61.29%. So the two percentage actually corresponds to the minima and maxima of the 12 percentage (3 pollutants *4 seasons).**

**In the revised manuscript, we have revised it to better explain what the two percentage mean.**

-The sentence: "If we simply consider the difference between qualitative output (just the dominant meteorological factor with the largest ρ) revealed at 3h and 24h scale to reveal the temporal effects of pollutant-meteorology association, the analysis was not complete." This sentence is not clear. From what can be deduced, the idea being conveyed is that because this is not sufficient, it is necessary to analyze the ρ, right?

**R: Thanks so much for this comment. Yes, you are right. This sentence mean, only know which meteorological factor had the largest ρ was just a qualitative conclusion, which cannot comprehensively reveal the difference of pollutant-meteorology association at 3h and 24h scales. Therefore, we should further analyze the detailed ρ for all meteorological factors, to present a quantitative and comprehensive comparison.**

**In the revised manuscript, we have revised it accordingly.**

-"Relatively 'mild' reactions," I believe "mild" is not precise; perhaps it would be necessary to replace the term or make it clear what is meant.

**R: Thanks so much for this comment. Here we would like to use the "mild" to show a difference to "intense" reactions. As you mentioned, this may not be accurate enough. And thus we prefer to change it to "less-intense" in the revised manuscript.**

-"The dominant meteorological factor for Northern China was mainly wind, especially during the heavily polluted winter." It's not clear to me if this is the case when looking at the graphs. What should be understood by "Northern China"? NW/N-Center/NE?

**R: Thanks so much for your comment. Yes, as you pointed here, we did not clearly and correctly explained that "Northeast China". We have revised this part according to your comment.**

**Again, thanks so much for pointing this out.**

-Line 228: Remove the "and."

**R: Thanks so much for pointing this out. This word has been deleted.**

-"The heavily polluted season for O3 and PM was winter and summer, respectively." I think the authors mean the opposite.

**R: Thanks so much for this comment. It's a very good suggestion and we have corrected it accordingly in the revised manuscript.**

At the end of line 336, add the year of the study.

**R: Thanks so much for this comment. It's a very good suggestion and we have corrected it accordingly in the revised manuscript.**

**List of all relevant changes made in the manuscript:**

Line 3: 'Miaoqing Xu[1]' → 'Miaoqing Xu[1,2]'. 'Manchun Li[2]' → 'Manchun Li[3]'. 'Bingbo Gao[3]' → 'Bingbo Gao[4]'

Line 5: 'College of Global and Earth System Sciences' → 'Faculty of Geographical Science'.

Line 7: '[2]Hubei Provincial Academy of Eco-environmental Sciences (Hubei Eco-environmental Engineering Assessment Center), Wuhan 430079, China' has been added to the revised manuscript.

Line 9: '[2]School' → '[3]School'.

Line 10: '[3]College' → '[4]College'.

Line 71: 'Per-3h' → '3 hourly'.

Line 139: 'For all three airborne pollutants, the dominant meteorological factor at the 3h and 24h scale was the same in only around 31.68% ~ 61.29%,' → 'As shown in Table 4, the consistence between dominant meteorological factors for PM2.5, PM10 and O3 at two temporal scales varied significantly (ranging from 31.68% ~ 61.29%)'.

Line 157: 'As one can see' → 'As can be seen'.

Line 170: 'In this case, if we simply consider the difference between qualitative output (just the dominant meteorological factor with the largest $\rho$) revealed at 3h and 24h scale to reveal the temporal effects of pollutant-meteorology association, the analysis was not complete. Therefore, we further presented the detailed comparison of the influence of individual meteorological factors on $O_3$, $PM_{2.5}$ and $PM_{10}$ at 3h and 24h respectively.' → 'In this case, if we simply consider the difference between the dominant meteorological factor (with the largest $\rho$) at 3h and 24h scale, the analysis was qualitative and not sufficient, which cannot comprehensively reveal the difference of pollutant-meteorology association at different temporal scales. Therefore, we further analyzed the detailed $\rho$ for all meteorological factors on $O_3$, $PM_{2.5}$ and $PM_{10}$ at two temporal scales respectively, to present a quantitative and comprehensive comparison.'.

Line 228: 'relatively mild' → 'less-intensive'.

Line 263: 'Northern' → 'Northeast'.

Line 264: 'and' has been removed.

Line 297: '$O_3$ and PM was winter and summer respectively' → '$O_3$ and PM was summer and winter respectively'.

Line 344: 'in 2020' has been added to the revised manuscript.

Line 350: 'relatively mild' → 'less-intensive'.